# Effects of population mobility on the COVID-19 spread in Brazil

**Eduarda T. C. Chagas**[1] ☵, **Pedro H. Barros**[1] ☵, **Isadora Cardoso-Pereira**[1], **Igor V. Ponte**[2], **Pablo Ximenes**[2,3], **Flávio Figueiredo**[1], **Fabricio Murai**[1], **Ana Paula Couto da Silva**[1], **Jussara M. Almeida**[1], **Antonio A. F. Loureiro**[1], **Heitor S. Ramos**[1] *

1 Department of Computer Science, Federal University of Minas Gerais, Belo Horizonte, Minas Gerais, Brazil, 2 Department of Motor Vehicles, Government of the State of Ceará, Fortaleza, Ceará, Brazil, 3 School of Cybersecurity and Privacy, College of Computing, Georgia Institute of Technology, Atlanta, Georgia, United States of America

☵ These authors contributed equally to this work.
* ramosh@dcc.ufmg.br

## Abstract

This article proposes a study of the SARS-CoV-2 virus spread and the efficacy of public policies in Brazil. Using both aggregated (from large Internet companies) and fine-grained (from Departments of Motor Vehicles) mobility data sources, our work sheds light on the effect of mobility on the pandemic situation in the Brazilian territory. Our main contribution is to show how mobility data, particularly fine-grained ones, can offer valuable insights into virus propagation. For this, we propose a modification in the SENUR model to add mobility information, evaluating different data availability scenarios (different information granularities), and finally, we carry out simulations to evaluate possible public policies. In particular, we conduct a case study that shows, through simulations of hypothetical scenarios, that the contagion curve in several Brazilian cities could have been milder if the government had imposed mobility restrictions soon after reporting the first case. Our results also show that if the government had not taken any action and the only safety measure taken was the population's voluntary isolation (out of fear), the time until the contagion peak for the first wave would have been postponed, but its value would more than double.

**Data Availability Statement:** Data cannot be shared publicly due to privacy concerns. Data are available from the Department of Motor Vehicles, Government of the State of Ceara, Fortaleza, Ceara, Brazil Institutional Data Access / Ethics Committee

## Introduction

In December 2019, a new virus from the coronavirus family, SARS-CoV-2, was reported in China. The virus, which is responsible for the COVID-19 disease, quickly spread across boundaries, affecting the whole world, and has become one of the most significant health challenges of the 21st century. A little more than a year later, in April 2021, 132 million cases and 2,8 million deaths have been officially reported worldwide [1]. While most of the world's population are still in the vaccination process, global health experts expect more cases and deaths in subsequent months. Brazil accounts for a large share of cases worldwide (it is about 20% on August 6th, 2021) according to data from state governments [2].

(contact via Ilton.junior@detran.ce.gov.br) for researchers who meet the criteria for access to confidential data. The data underlying the results presented in the study are available from (Ilton Júnior <Ilton.junior@detran.ce.gov.br>).

**Funding:** HSR acknowledges funding support from the Pró-Reitoria de Pesquisa (PRPq) of the Universidade Federal de Minas Gerais (UFMG) for the publication costs. HSR is funded by grant #2020/05121-4, São Paulo Research Foundation (FAPESP). AFFL is funded by grant #2018/23064-8, São Paulo Research Foundation (FAPESP). HSR is funded by grant 311750/2018-4, Conselho Nacional de Desenvolvimento Científico e Tecnológico, CNPq. ETCC and ICP hold scholarships from the Coordenação de Aperfeiçoamento de Pessoal de Nível Superior, CAPES. PHB holds a scholarship from the Conselho Nacional de Desenvolvimento Científico e Tecnológico, CNPq. FM is funded by grant APQ-02337-21, Fundação de Amparo à Pesquisa do Estado de Minas Gerais, FAPEMIG. The funders had no role in study design, data collection and analysis, decision to publish, or preparation of the manuscript.

**Competing interests:** NO authors have competing interests.

Since transmission occurs through direct contact between people, social isolation [3] is a primary measure in combating virus dissemination. Therefore, a lower population mobility may yield to a lesser spread of the disease in the community. Considering population mobility as an essential factor for contagion spread, modeling the pandemic dynamics through mobility data has been shown to be efficient and provide valuable insights to guide public service policies. For example, one of the earliest developments in this direction was performed by Kraemer *et al.* [4], who discussed the impact of travel restrictions in mainland China. Such restrictions, alongside measures of social distancing and quarantine, have rapidly decreased the force of infection and hence controlled the disease spread. Using human mobility data, the authors observed this change immediately after an intervention by measuring the correlation between the mobility indexes and the growth rate of the disease. In a similar approach, Chinazzi *et al.* [5] also evaluated the impact of travel restrictions on the growth rate of the COVID-19 disease in the Wuhan region, in China. Additionally, Du *et al.* [6] estimated the probability of transportation of COVID-19 from Wuhan to other cities in China before the first quarantine. These studies reached similar conclusions: restricting mobility between China provinces has drastically reduced the virus spreading. In turn, Buckee *et al.* [7] discussed the importance of aggregating distinct data from multiple sources to monitor social distancing interventions, building reliable information in space and time and reflecting an approximation of population-level mobility rather than individual patterns.

The studies of the spread of COVID-19 often rely on epidemiological compartmental models [8–10], such as the susceptible-exposed-infected-recovered (SEIR) model [11]. In these models, a fixed population of individuals is divided into four different states (or compartments) according to the following dynamics. All individuals who have not had contact with the virus are in the *susceptible* state. The individuals in the *exposed* state are those who have been infected, based on the force of infection, but have not yet become infectious themselves due to the incubation period. After that period, those individuals become *infected*, moving to the corresponding state. The *recovery* state covers all post-infection scenarios, i.e., the individuals in this state may have recovered (and are unlikely to be reinfected) or died.

The COVID-19 spread was modeled by variations of the SEIR model as well. Li *et al.* [12] used the Susceptible-Exposed-Infected-Confirmed-Removed (SEIQR) model with data about patients that are laboratory-confirmed to show that an earlier lockdown in Wuhan would have worsened the outbreak in the city but would have helped the rest of the world.

Davies *et al.* [13] modified the Susceptible-Exposed-Notified-Underreported-Removed (SENUR) model to account for age-stratified transmission. The authors used the reported cases and the age distribution of patients from different countries to concluded that regions with an older population are likely to have more COVID-19 cases.

Specifically focused on Brazil, several studies investigated the impact of human mobility on the spread of the disease using different types of data. For instance, some studies [14, 15] used Brazilian census information about people and terrestrial vehicles as well as air transportation data, basing the measurement of SARS-CoV-2 spreading patterns on data collected before the pandemic. Other studies calculated such spreading patterns between cities through mobile phone data [16–18]. Serafino *et al.* [19] implemented a protocol for optimized quarantines based on the analysis of contact tracking networks, in order to dismantle the coronavirus transmission chain with the minimum necessary interruptions. To monitor the evolution of the transmission contact network before and after quarantines, a compilation of hundreds of human mobility applications deployed in Latin America was used.

Inspired by this body of work, our objective is to investigate mobility's impact on the SARS-CoV-2 spread. To this, we modify the SENUR model to use mobility as part of its input,

allowing us to study the effect of this new variable on the disease spread through simulated scenarios. To that end, we use two types of data:

1. Mobility data captured by vehicles before and during the pandemic outbreak;

2. Epidemiological data about the virus containing information such as the date of onset of symptoms and the test performed.

Our main contribution in this work is to show how mobility data in different granularity along with epidemiological data can offer valuable insights into virus propagation. Although epidemiological compartmental models can satisfactorily model the pandemic behavior, they cannot answer questions regarding the impact of mobility, such as quantifying how the government measures of mobility restriction affect the infection rate, which we investigate here. Hence, we focus on answer the following questions:

**How does people's mobility affect the pandemic in Brazil?** Here we seek to understand the impact of mobility on the virus spread and, consequently, to measure this impact on the numbers of infected individuals and deaths. To address this question, we analyze the effect of mobility under the following different pandemic scenarios: 1) the individuals do not change their mobility behavior, and the government does not restrict it in any way. In other words, the mobility is stable during the pandemic; 2) the government enforces a lockdown (and people adhere) right after the confirmation of the first case.

**Which benefits do different mobility data granularities bring to the study of the pandemic effects?** To answer this question, we evaluate mobility data in two different granularities: city-level (or coarse) and neighborhood-level (or fine) granularity. For the former, we are using the aggregated time-series data provided by Waze reports. For the latter, we use a flow matrix obtained from Automatic License Plate Readers (ALPR) from the mobility data of the State of Ceará Department of Motor Vehicles (DETRAN-CE). We emphasize that, although fine granularity data can provide more details about urban mobility, potentially allowing more precise models, they are harder to obtain because they come from public policy actions, such as traffic department road-level data. Hence, it is important to study other granularities as well to assess the trade-offs between them.

In our results, we observed that models estimated with data from more/less restrictive public policies present a lower/higher number of notified infected individuals when compared to the model fit to real data. We noticed that the mobility factor of our epidemiological model tends to capture the trend of the restriction measures applied by the governments. We also observe that our model estimates a mobility quantifier, and it presents coherent results for the cities analyzed regardless of the granularity of the mobility data used. Note that, with fine-grained data, we can estimate local parameters that enable the design of public policies tailored for each city region. Instead, for coarse-grained data, public polices will affect the whole city.

The remainder of this paper is organized as follows. In the next section, we present in detail the epidemiological and mobility data used throughout this work. Next, we discuss the applied methodology, which includes the proposed compartmental model variant. The following section presents our analyses and main results. Finally, we conclude our findings and discuss future directions.

## Data sources

This work explores data from five major Brazilian cities that have undergone mobility restriction policies in 2020, namely: Fortaleza, Belo Horizonte, Porto Alegre, Rio de Janeiro, and São

Paulo. These cities were select because of the diversity in terms of demographics as well as political and social context within Brazil. Specifically, São Paulo, Rio de Janeiro, and Belo Horizonte are the most populous metropolitan regions in Brazil. Moreover, in Brazil, the Southern region was the most impacted by the H1N1 virus pandemic, commonly called Swine Flu [20]. Hence we chose a large city in that region (Porto Alegre) to assess how it was affected by the new Coronavirus pandemic. Finally, thanks to an agreement between institutions, we obtained data regarding the mobility of cars in Fortaleza, a large city in the Northeast of Brazil, allowing us to perform a finer granularity analysis, as described below.

Specifically, recall that we here propose to use o mobility data at two different granularities to study the COVID-19 spread, namely city and neighborhood-level granularity. For the study at the city level, for all five cities, we use data collected by Waze and available in its Mobility Report [21], from 02/25/2020 to 08/09/2020 (totaling 168 days). The data released by the mobile application corresponds to the percentage of variation (aggregated and anonymized) in the total distance traveled compared to a baseline. The baseline is defined by the Waze and corresponds to the average value calculated for each corresponding day of the week concerning the period from 02/11/2020 to 02/25/2020 (the pre-pandemic period in Brazil).

To capture the mobility between different neighborhoods of a given city, we use data collected by the Government of the DETRAN-CE. This data contains anonymized readings from 265 Automatic License Plate Readers (ALPR) in 52 neighborhoods in the city of Fortaleza, from 01/16/2020 to 05/04/2020 (totaling 110 days). The anonymization procedures took place in DETRAN-CE before researchers had direct access to the data. In addition, DETRAN-CE employed suppression-based anonymization techniques by removing all vehicle-specific data and replacing it with individual randomized tags [22]. This way, the resulting anonymization process aimed to balance the dataset's utility and privacy protections while prioritizing utility. Besides all technical safeguards, researchers that had any direct involvement with the dataset have signed a data-sharing agreement that, among other things, grants legal standing and protection for several ethical considerations.

However, DETRAN-CE's dataset presents limitations, primarily due to its incomplete spatial coverage of the city, since, as said before, the locations of the ALPR units do not favor widespread tracking. Nevertheless, we assume that such data constitute a representative sample of vehicular mobility behavior. In fact, through the Granger causality test [23], we show that they are suitable to determine the strength of viral infection in a city (Section *Case study II: Fine-grained analysis*). Besides, vehicle mobility can represent various individuals traveling in predefined routes (such as buses). Hence, it does not consider the trip's intention (e.g., taxis usually travel when they have passengers, without a predefined travel routine). Moreover, as such data generally present sensitive information, those datasets are commonly private and difficult to obtain.

Moreover, we correlate the traffic information captured by DETRAN-CE's ALPR data with mobility indices provided by Google's COVID-19 Community Mobility Reports [24]. This report presents trends in people's movement over time in geographic space (such as neighborhoods), in different categories of locations, such as retail and recreation, supermarkets and pharmacies, parks, transit stations, workplaces, and homes. Hence, this analysis allows us to understand the activities captured by the traffic data, i.e., which activities people in Fortaleza prefer to use vehicles to access.

Furthermore, we obtained statistics about the COVID-19 scenario in each city from two different platforms of the Brazilian government:

1. Coronavirus Panel [2], which is the official disclosure channel used by the government and provides daily statistics of cases and deaths, and

2. Opendata SUS [25], a smaller dataset that includes more details on clinical (e.g., the date of the first symptom presented by the patient and the patient's clinical evolution) and demographic (e.g., the patient's residence) information of the cases. This second dataset allows the estimation of the delay distribution between symptom onset and case notification.

Finally, we aggregate the city's neighborhoods in Fortaleza into sub-regions since it reduces the number of variables of our method, increasing the model's accuracy. We base the construction of these sub-regions on geographic connectivity and the human development index (HDI). In Section *Case study II: Fine-grained analysis* we analyze the relationship between the HDI of neighborhoods in Fortaleza and the recovery from COVID-19. To this study, we use data obtained by the study carried out by the Fortaleza Municipal Secretaria for Economic Development (SDE) [26]. These results used data from the last Brazil Demographic Census carried out in 2010 as a basis.

## Methodology

### Estimating the delays between case onset and report

We assume that a person can infect others from the symptom onset. However, this date generally is delayed from the officially notified date for several reasons, such as difficulties inserting the record in the system and delays in medical exams (collection and results). In any case, we estimate such time delay between the officially notified date (in the Coronavirus Panel) and symptom onset dates to obtain more reliable results, using the Opendata SUS platform dataset. This dataset contains a subset of Brazil's COVID-19 official cases, with Coronavirus Panel being the complete set. However, the Opendata SUS platform provides both symptom onset and the official registration dates. Following the methodology used by Abbott et. al. [27], we performed a sampling process to estimate the probability distribution of the delay in the officially notified cases based on the Opendata SUS platform. Next, we projected the actual transmission date for each infected individual presented in the Coronavirus Panel data. This process consisted of sampling possible values for the delay and adjusting these samples through distributions, described in detail in the following.

The symptom onset data were adjusted using the Gamma and Exponential distributions with the statistical modeling program Stan [28]. We selected the best fit obtained among the chosen distributions through the approximate criterion of leave-one-out cross-validation (LOOCV) [29]. Hence, the distribution model most appropriate was the one with the lowest LOOCV value (Gamma or Exponential). Finally, with the parameters of such distribution, we rewind the dataset by resampling the data using bootstrap.

Thus, based on the empirical delay distribution, we could sample the expected delay values and perform a temporal rewind of the Coronavirus Panel data. Fig 1 shows, for the Fortaleza city, the empirical delay distribution, calculated using Opendata SUS data and the estimated distribution of the delay (estimated by the Stan software). The vertical lines represent the mean of the delay value in each distribution. Additionally, it also shows the distribution of notified cases before and after data rewind. The data interval for all plots is from 03/20/2020 to 09/30/2020.

We can see that rewinding data make it smoother. Before this pre-processing step, the data presents many peaks, as we can see in the third plot (when looking from left to right). It is likely to represent accumulated data delayed for some unknown reason, such as data collected during weekends. Differently, we observed after pre-processing the gradual spread of the virus, representing a more realistic distribution of the number of cases.

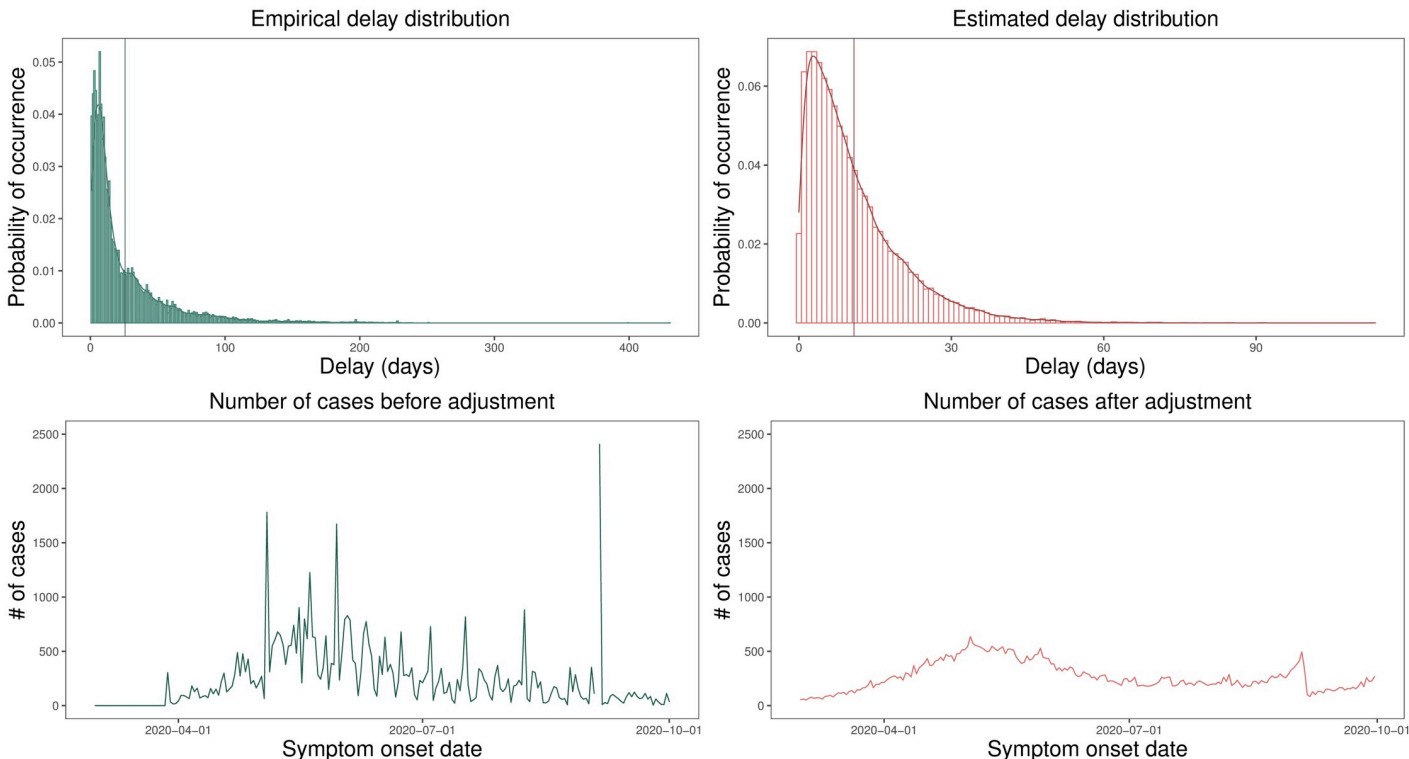

**Fig 1. Probability distribution of the delay in the notified cases.** The distributions of delay and number of cases estimated from the data notified by Opendata SUS platform and the Coronavirus Panel before and after adjustments of the lag between symptom onset and official notification in Fortaleza/CE (Brazil). From right to left, we have: empirical delay distribution (with $\mu = 25.72$ and $\sigma = 31.85$), estimated delay distribution(with $\mu = 10.85$ and $\sigma = 9.61$), cases distribution before fits, cases distribution after fits. In addition, $\mu$ is the mean of the distribution and $\sigma$ is the standard deviation and the vertical lines on the first and second plots represent the mean of the delay value in each analyzed distribution.

## Reproduction number

A fundamental question for public policy response during a pandemic is quantifying the transmissibility of the infectious disease. We calculate this measurement through the Reproduction Number $R(t)$, which is the average number of cases that a single infected person transmits, i.e., how many people a single individual could infect.

We based the calculation of $R(t)$ in this work on Abbott et al. [27], which uses the Markov Chain Monte Carlo (MCMC) [30] to quantify the pandemic's spread uncertainty from all inputs into the final parameter estimation. Note that the number of onset cases in a day is defined as a random variable after rewinding.

## SENUR model equipped with mobility information

Let $N_i$ be the population of the region $i$ in a city. For a given region $i$ and time $t$, we have the following states:

- $S_i(t)$: the number of susceptible individuals. These are the healthy individuals whom the disease can firstly infect.

- $E_i(t)$: the number of exposed individuals, e.g., those who have been infected but are not yet infectious (the disease is in the incubation period).

- $I_{N_i}(t)$: the number of (notified) clinical individuals. These individuals have been infected and reported their cases to the health authorities (for instance, going to a hospital).

- $I_{U_i}(t)$ is the number of underreported infected individuals. In this compartment, the individuals have been infected, but they have not reported their cases.

- $RE_i(t)$: the number of removed individuals. Such individuals have been infected and either recovered or died from the disease.

It is worth noting that both $I_{N_i}(t)$ and $I_{U_i}(t)$ states contain individuals who can transmit the disease, i.e., infect susceptible individuals. The disease dynamics, that change an individual from a state to another, is described as follows:

1. Each healthy individual can become contaminated in their region of origin, through local transmission, or move to other geographically connected locations and be vulnerable to global transmission. The infection rate of a healthy individual (residing in the $i$ region) at time $t$ is given by the force of infection $\lambda_i(t)$.

2. A healthy individual, infected with a $\lambda_i(t)$ rate, moves from the susceptible to the exposed compartment.

3. Once exposed, the individual has a $y_i(t)$ probability of presenting symptoms, looking for a hospital, having access to care, and being notified, thus going to the clinically infected state (notified); otherwise, they may not be reported (e.g., asymptomatic cases), going to the sub-clinically infected state (underreported). We consider that both clinical and subclinical individuals have the same transmission rate.

4. After being infected, the individual is moved to the removed compartment. We consider it as the last stage in the spread of an infectious disease. Although we have scientific evidence that possible reinfections may occur, for simplicity, we assume in our modeling that when in the removed compartment, the individual is immune (i.e., cannot be re-infected), being removed from the system. Moreover, our proposal considers a constant population, ignoring demographics. We also do not consider the arrival of individuals.

Fig 2 summarizes the aforementioned dynamics. Additionally, we can describe the dynamics of the SENUR model used in this work through the set of the following equations:

$$\frac{dS_i(t)}{dt} = -\lambda_i(t)S_i(t), \tag{1}$$

$$\frac{dE_i(t)}{dt} = \lambda_i(t)S_i(t) - y_i(t)d_E^{-1}E_i(t) - [1 - y_i(t)]d_E^{-1}E_i(t), \tag{2}$$

$$\frac{dI_{N_i}(t)}{dt} = y_i(t)d_E^{-1}E_i - I_{N_i}(t)d_N^{-1}, \tag{3}$$

$$\frac{dI_{U_i}(t)}{dt} = [1 - y_i(t)]d_E^{-1}E_i(t) - I_{U_i}(t)d_U^{-1}, \tag{4}$$

$$\frac{dRE_i(t)}{dt} = d_N^{-1}I_{N_i}(t) + I_{U_i}(t)d_U^{-1}, \tag{5}$$

where $d_E^{-1}$ represents the transition rate from exposed individuals $E_i(t)$ to infected individuals $I_{N_i}(t)$ and $I_{U_i}(t)$; $d_N^{-1}$ represents the transition rate from notified infected individuals $I_{N_i}(t)$ to

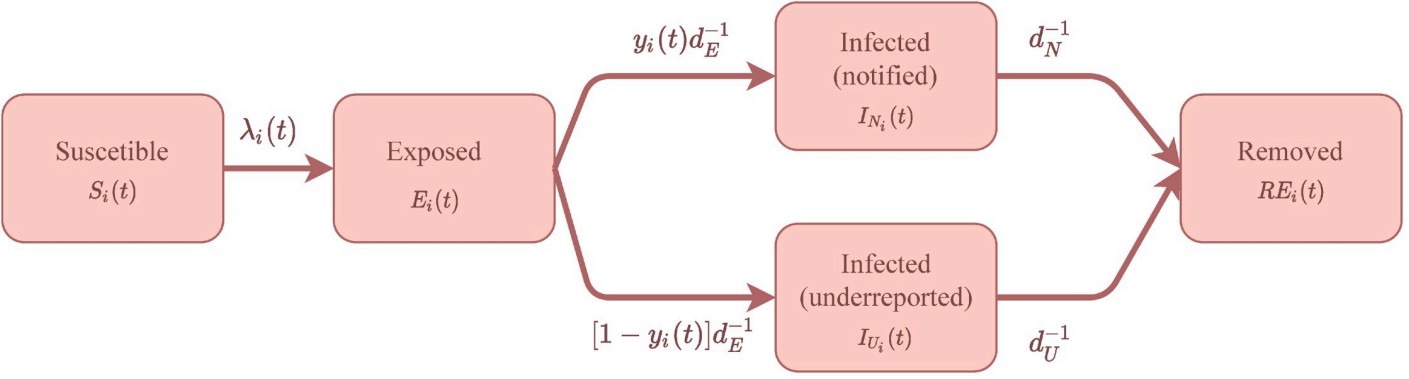

**Fig 2. SENUR model.** Schematic for the SENUR model used in this work.

removed individuals $RE_i(t)$; and $d_U^{-1}$ represents the transition rate from underreported infected individuals $I_{U_i}(t)$ to the removed individuals $RE_i(t)$.

Although SENUR models can satisfactorily model the pandemic behavior, they cannot answer questions regarding the impact of mobility. For instance, we cannot quantify the decrease in mobility that may occur when the government decrees measures of mobility restriction. Therefore, we propose in this work an extension of this model to investigate human mobility's influence on the spread of the COVID-19 pandemic.

We assume that the element $C_{ij}$ represents the city's daily mobility. The impact of mobility on the virus spread at time $t$ is given by the matrix $W_{ij}(t) = q_t C_{ij}$, where $q_t \in [0, 1]$ is a scalar, time-dependent parameter estimated by the model to quantify the influence of mobility on pandemic dynamics. Hence, we model the virus transmission by capturing people's mobility as they move from one neighborhood to another and exploring its relationship to the spread of the disease. Thus, for this work, we know a prior that mobility before the periods of isolation measures are more significant than the period analyzed in the article, thus justifying the choice of $0 \leq q_t \leq 1$. However, we can adopt a less restrictive range without losing the generalizability of the technique. So, the model estimates a $q_t$ for this period, a value similar to that found in the experiment where $0 \leq q_t \leq 1$ (0.381).

Moreover, we consider that the mobility parameter $q_t$ changes according to government policies regarding social distance. Thus, $q_0 = 1$ when the mobility is as usual. In other words, $q_0 = 1$ in the data available before the first notified case of COVID-19. We calculate the $q_t$ values as a proportion of the normal mobility. The simulation's details are in section Model Solving.

The probability of a healthy individual resident of cluster $i$ becoming infected, that is, the force of infection of cluster $i$ considering local and global transmission factors, is

$$\lambda_i(t) = \mu_i \times \sum_j \left[ W_{ij}(t) \times \left( \frac{I_{N_j}(t) + I_{U_j}(t)}{N_i} \right) \right],$$

where $\mu_i$ is a regularization term that defines how nonpharmaceutical interventions (NPIs) affect the virus transmissibility rate in the cluster $i$, i.e., $\mu_i$ models the actual virus transmissibility rate between the susceptible individuals from $i$ ($N_i$) and the infected individuals from $j$ ($I_{N_j}(t) + I_{U_j}(t)$). For instance, if the population uses masks and applies other NPI measures, the $\mu_i$ value must decrease.

This model can also be applied to an index when using only the region. When using the mobility index, we consider all the paths of individuals in the analyzed region, regardless of their origin or destination (coarser spatial granularity). So, we can simplifies the force of infection equation to:

$$\lambda(t) = \mu_i \times W(t) \times \left(\frac{I_N(t) + I_U(t)}{N}\right).$$

The granularity of the resulting model is directly associated with the data granularity. Thus, if we only have mobility information about one region, we can only use our model to make inferences about this single region. Otherwise, if the mobility data contain information that break downs the region into smaller areas (e.g., neighborhoods in a city), we can make inferences at a finer granularity.

An overview of the methodology applied in this article is shown in Fig 3.

The first step is the rewind data process, which consists of estimating the delay between case onset and report. In step 2, we estimate the reproduction number $R(t)$ and check the Granger-causality with the mobility data.

If the mobility intensity, represented by the number of vehicles observed, Granger-causes $R(t)$, we proceed with the modeling process. In Step 3, we use our SENUR adapted model, which includes the mobility information, to estimate all parameters. After those steps, our model is ready to be used.

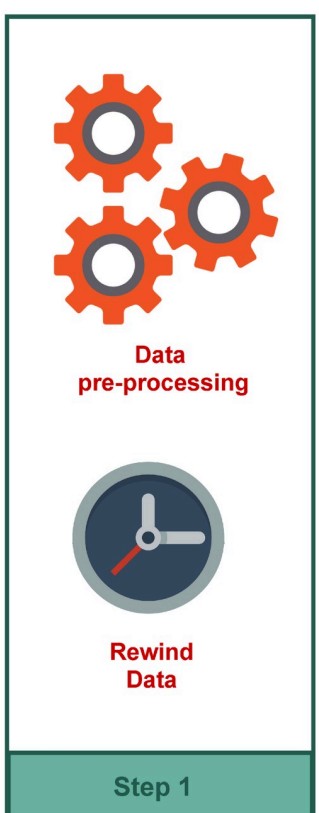
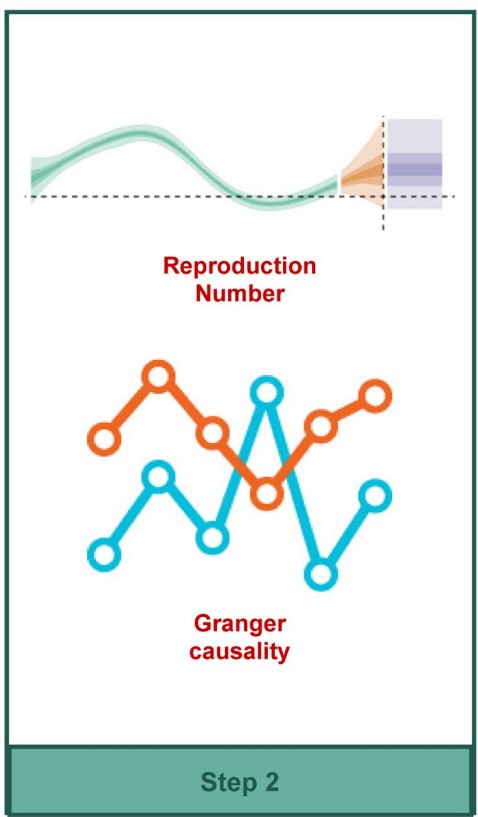
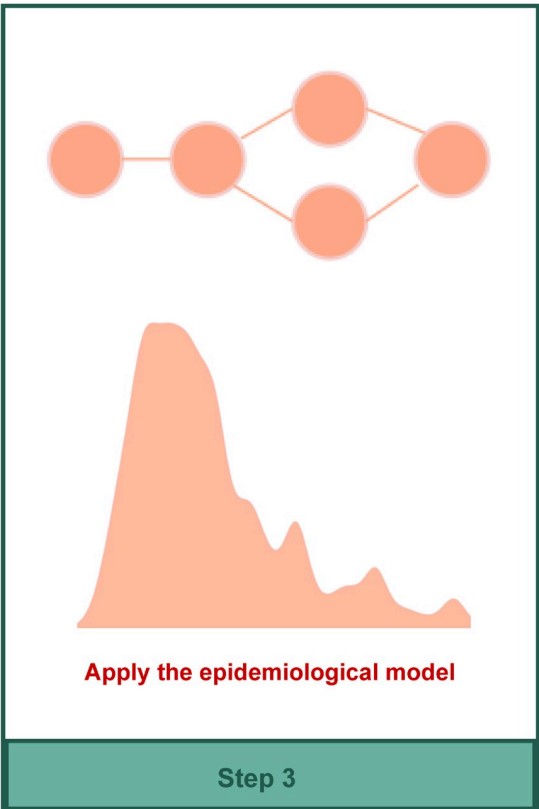

**Fig 3. Methodology scheme.** Overview of the methodology used for the evaluation of the mobility effects on transmission and control of COVID-19.

## Model solving

To solve the described model, we used the Bayesian inference based on a sampling distribution via MCMC. Let $\Pr(\mathcal{X}|\Theta)$ be the probability of a model with parameters $\Theta$ generate data $\mathcal{X}$. Hence, the Bayesian framework can characterize the posterior distribution as $\Pr(\Theta|\mathcal{X})$, that represents the probability that $\mathcal{X}$ data was generated by a model with parameters $\Theta$ [31]. By Bayes' rule, we have $\Pr(\Theta|\mathcal{X}) \propto Pr(\mathcal{X}|\Theta) \Pr(\Theta)$, where $\Pr(\Theta)$ is the prior distribution of the model with parameters $\Theta$, containing the information about the parameters before observing the data.

Therefore, for transmission parameters $\Theta = \{\lambda_i(t), y_i(t), d_E^{-1}, d_U^{-1}, d_N^{-1}\}$ and initial constraints, a compartmental model defines a solution for each compartment. In this context, we want to associate the solution generated by the model with the observed data, i.e., the infected individuals reported by the health organizations each day. We consider that $I_{N_i}(t)$, representing these notified cases, is a counting distribution (Negative Binomial). This distribution allows us to use $I_{N_i}(t)$ as a random variable and account for over-dispersion through the parameter $\varphi$, as $\mathcal{X}(t) \sim NegBin(S(t), \phi)$, where $S(t) = \sum_{i=1}^{N} I_{N_i}(t)$.

As described before, we estimate the empirical distribution of the delay, i.e., the time for an exposed individual to present the first symptoms. Thus, we use the empirical distribution to estimate the number of individuals exposed (and the date of exposure).

We started our simulations by scattering the first exposed individuals randomly in the analyzed regions. We used as a prior distribution of our analyzed variables $d_E^{-1} \sim \Gamma(3,4)$, $d_N^{-1} \sim \Gamma(2.1, 4)$, $d_U^{-1} \sim \Gamma(3.2, 3.7)$, $y_i(t) \sim \mathcal{U}(0, 1)$, where $\Gamma$ and $\mathcal{U}$ denote the Gamma and Uniform distributions, respectively. We used these models in accordance to [13, 32, 33]. For the parameter of mobility influence $q_t$, we only consider that the value was in the intervals $[0, 1]$, and therefore, use $q_t \sim \mathcal{U}(0.1)$ as a prior distribution. We performed 4000 repetitions of the simulation (through the MCMC) to estimate the model's parameters distribution.

To summarize, a Bayesian model couples a mathematical model of what we know about the parameters in the form of a prior and a sampling distribution. We implement it using PyStan [34], which consists of a Python library to use the Stan language.

## Results and discussion

### Case study I: Coarse-grained analysis

To evaluate the model, we use the mobility data released by the Waze platform. It contains the overall distance traveled by vehicles in a city relative to a baseline calculated in a period before the pandemic. Therefore, this dataset only allows a coarse-grained analysis since we do not have information about the mobility among the regions of a city.

Our model represents human mobility by the parameter $q_t$, estimated for each time $t$ via MCMC. Table 1 shows how we assign $q_t$ to different periods in all cities studied. For instance, for Fortaleza, $q_0$ corresponds to the period before $FO_0$ (the first notified case), $q_1$ to the period before $FO_1$ (trade closure), and so forth. Initially, we noticed that all cities analyzed had reduced mobility with the appearance of the first quoted case of COVID-19. This behavior is expected because people tend to naturally reduced mobility due to the fear of contagion after the appearance of the first notified case. Furthermore, we observe that São Paulo has the highest $q_1$ mean, probably because it was the first case reported in Brazil.

We observed some general behaviors for the cities analyzed. For example, after adopting trade closure measures, the $q_t$ average showed a reduction, thus showing evidence about the effectiveness of this measure. It is worth noting that we do not have data about the control case, i.e., how does the pandemic behave if the trade was not closed? However, we only have

                                   

**Table 1. Periods used to estimate the mobility parameter $q_t$ for different cities.** The table presents the periods used in our model to estimate the different values of the mobility parameter $q_t$ alongside the $q_t$ sample mean. We collected these periods from news provided by the government of each city in 2020. The description describes the main events related to the period.

| Start | Finish | Label | q | Description |
|---|---|---|---|---|
| **Fortaleza** | | | | |
| – | 03/15 | – | $q_0 = 1.000$ | Before first notified case of COVID-19 at Fortaleza |
| 03/15 | 03/20 | $FO_0$ | $q_1 = 0.417$ | First notified case of COVID-19 at Fortaleza |
| 03/20 | 05/05 | $FO_1$ | $q_2 = 0.009$ | Trade closure |
| 05/05 | 05/30 | $FO_2$ | $q_3 = 0.121$ | Lockdown |
| 05/30 | 06/08 | $FO_3$ | $q_4 = 0.232$ | Reopening of activities (work and trade) |
| 06/08 | 07/27 | $FO_4$ | $q_5 = 0.206$ | Reopening of trade |
| 07/27 | 09/01 | $FO_5$ | $q_6 = 0.355$ | Return of public transport |
| 09/01 | 09/20 | $FO_6$ | $q_7 = 0.612$ | Reopening of bars |
| **Belo Horizonte** | | | | |
| – | 03/16 | $BH_0$ | $q_0 = 1.000$ | Before first notified case of COVID-19 at Belo Horizonte |
| 03/16 | 03/18 | $BH_1$ | $q_1 = 0.352$ | First notified case of COVID-19 at Belo Horizonte |
| 03/18 | 04/06 | $BH_2$ | $q_2 = 0.253$ | Decree defining agglomeration closure and public events) |
| 04/06 | 04/16 | $BH_3$ | $q_3 = 0.227$ | Decree defining the closure of trade |
| 04/16 | 05/22 | $BH_4$ | $q_4 = 0.266$ | Mandatory use of masks in public spaces |
| 05/22 | 06/26 | $BH_5$ | $q_5 = 0.345$ | Preventive measures in public transport |
| 06/26 | 07/24 | $BH_6$ | $q_6 = 0.303$ | Suspend some preventive measures to return activities |
| 07/24 | 08/04 | $BH_7$ | $q_7 = 0.432$ | Reopening of football matches (without fans in stadiums) |
| 08/04 | 09/13 | $BH_8$ | $q_8 = 0.447$ | Reopening of trade |
| **Porto Alegre** | | | | |
| – | 03/16 | $PA_0$ | $q_0 = 1.000$ | Before first notified case of COVID-19 at Porto Alegre |
| 03/11 | 03/17 | $PA_1$ | $q_1 = 0.638$ | First notified case of COVID-19 at Porto Alegre |
| 03/17 | 04/22 | $PA_2$ | $q_2 = 0.403$ | Trade closure and cancellation of major events |
| 04/22 | 05/03 | $PA_3$ | $q_3 = 0.585$ | Reopening of civil construction trade activities |
| 05/03 | 05/20 | $PA_4$ | $q_4 = 0.423$ | Reopening of some strategic commercial sectors |
| 05/20 | 06/15 | $PA_5$ | $q_5 = 0.417$ | Reopening of trade |
| 06/15 | 07/04 | $PA_6$ | $q_6 = 0.252$ | Re-closing of large-scale activities |
| 07/04 | 08/06 | $PA_7$ | $q_7 = 0.121$ | Closure of amusement parks and beaches |
| 08/06 | 08/20 | $PA_8$ | $q_8 = 0.206$ | Reopening of outdoor fairs and beauty salons |
| 08/20 | 09/25 | $PA_9$ | $q_9 = 0.237$ | Reopening of shopping malls and shopping centers |
| **Rio de Janeiro** | | | | |
| – | 03/05 | $RJ_0$ | $q_0 = 1.000$ | Before first notified case of COVID-19 at Rio de Janeiro |
| 03/05 | 03/20 | $RJ_1$ | $q_1 = 0.598$ | First notified case of COVID-19 at Rio de Janeiro |
| 03/20 | 03/24 | $RJ_2$ | $q_2 = 0.447$ | Closure of intercity public transport |
| 03/24 | 04/18 | $RJ_3$ | $q_3 = 0.343$ | Closing of trade |
| 04/18 | 05/06 | $RJ_4$ | $q_4 = 0.242$ | Mandatory use of masks |
| 05/06 | 05/24 | $RJ_5$ | $q_5 = 0.201$ | Partial lockdown |
| 05/24 | 06/27 | $RJ_6$ | $q_6 = 0.144$ | Suspension of classes |
| 06/27 | 08/12 | $RJ_7$ | $q_7 = 0.248$ | Reopening of trade |
| 08/12 | 09/09 | $RJ_8$ | $q_8 = 0.247$ | Reopening of small and medium-sized events |
| 09/09 | 09/13 | $RJ_9$ | $q_9 = 0.316$ | Return of intercity public transport |
| **São Paulo** | | | | |
| – | 02/26 | $SP_0$ | $q_0 = 1.000$ | Before first notified case of COVID-19 at São Paulo |
| 02/26 | 03/14 | $SP_1$ | $q_1 = 0.913$ | First notified case of COVID-19 at São Paulo |
| 03/14 | 03/20 | $SP_2$ | $q_2 = 0.795$ | Suspension of medium and large events |
| 03/20 | 03/24 | $SP_3$ | $q_3 = 0.562$ | Suspension of public transport |

(*Continued*)

**Table 1.** (Continued)

| Start | Finish | Label | q | Description |
|---|---|---|---|---|
| 03/24 | 05/27 | $SP_4$ | $q_4 = 0.346$ | Trade closure |
| 05/27 | 05/31 | $SP_5$ | $q_5 = 0.252$ | Presentation of the reopening plan |
| 05/31 | 06/06 | $SP_6$ | $q_6 = 0.551$ | End of partial lockdown |
| 06/06 | 09/20 | $SP_7$ | $q_7 = 0.567$ | Reopening of bars |

evidences, in this case it is correlation, about the effectiveness of the measures. We are not able to make any causal inference to state that the trade closure indeed yields to a decrease of the pandemic.

We also see some interesting individual behaviors. Porto Alegre had two periods of intervention. At first, we noticed that in the reopening of civil construction activities, the city had a $q_4 = 0.423$. However, in the second moment, the intervention measure proved to be much more effective, indicating $q_7 = 0.121$, corresponding to the period of closing of parks and beaches. Furthermore, we see that Porto Alegre has the lowest mobility index value at the end of our analysis ($q_9 = 0.237$). This number shows evidence that as Porto Alegre was heavily affected by the H1N1 previous pandemic, the population may be more alert to the new pandemic (COVID-19), and thus, reacts earlier.

The cities of São Paulo and Rio de Janeiro reopened bars and restaurants and we observed an increase of $q_t$ after the adoption of these measures.

Fig 4 shows the estimated values of $q1$ and $q2$ using MCMC and their resulting distributions. Fig 4(a) and 4(b) show, respectively, the sampling distribution of $q_1$ and $q_2$ parameters for Fortaleza. We note that $q_1$ average is higher than $q_2$. This behavior is probably due to citizens following social distancing restrictions ahead of government actions when they notice the pandemic starting. Hence, with citizens staying in their homes, mobility decreased, and, consequently, there is a lower spread of the virus through mobility.

Given the inferred distributions of the mobility parameters $q_t$ for each period, we use our model to estimate the number of people infected over time. Fig 5 shows how the growth in the number of infected people has changed as the government imposed actions in an attempt to flatten the curve. Different colors represent the different periods used to estimate $q_t$, presented in Table 1. Shaded areas represent the confidence regions at 95% of significance. We observe that the proposed model can accurately capture the behavior of the curve representing the number of cases for all cities studied.

Furthermore, we use the inferred distribution of $q_t$ to make predictions for the 7-day window following the data used in the parameter estimation. The red dots represent this prediction in Fig 5. We consider a relatively short window since long-term predictions can become unreliable due to the dynamical behavior of the model's parameters. Even with the increase in uncertainty and, consequently, in the confidence regions, the data points are still within the expected values (i.e., within the confidence interval). Although compartmental models have an exponential characteristic, Fig 5 shows that we managed to keep the reported values within the confidence interval, including São Paulo, which is the city with the highest number of contagions registered in Brazil. Note that, in Fig 5, our model estimates the accumulated value of infected for each city. However, However, as shown in Fig 6, our model also captures daily cases.

Additionally, we used our model to simulate scenarios aiming to analyze the impact of mobility in the infection spread. Thus, we analyzed the pandemic behavior for two hypothetical scenarios of increasingly restrictive interventions, namely:

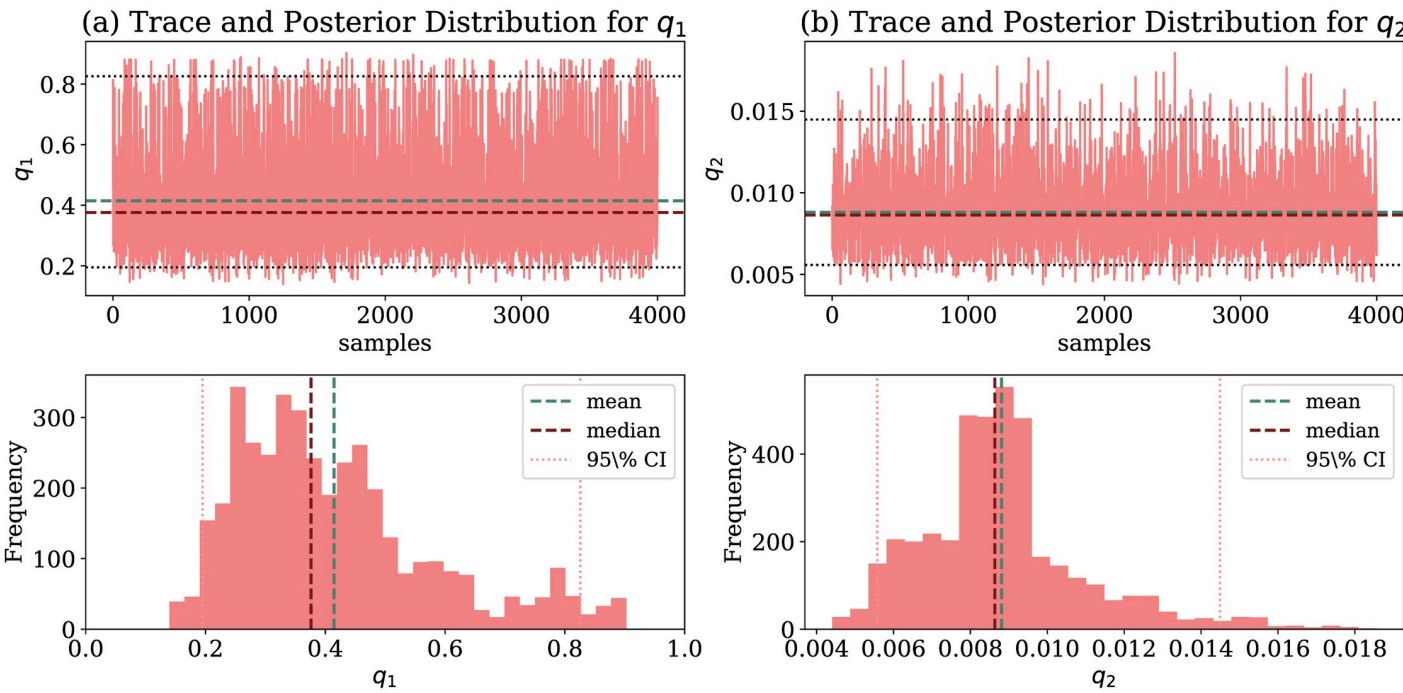

**Fig 4. Mobility parameters estimated by our model for Fortaleza.** Estimated values of mobility parameters $q_1$ and $q_2$ via MCMC and their resulting distributions. (a) $q_1$ (From 03/15 to 03/20). (b) $q_2$ (From 03/20 to 05/05).

**Scenario** I.   We assume that the government did not take any preventive measure, and the population did not change their mobility behavior. The value of the mobility coefficient is considered $q_0 = q_1 = 1$ during the whole period analyzed.

**Scenario** II.   We assume that, upon the occurrence of the first notified case, the government enforced the closing of the trade. We consider that each government of each analyzed city decided different times to this closing; hence, the value of $q_2$ is different for each city.

In these experiments, we analyze the ratio between the number of infected individuals under a given scenario and the number of infected individuals estimated by our model. Ratios smaller (resp. greater) than 1 represent a decrease (resp. increase) in the number of infected individuals under the hypothetical scenario relative to what took place in reality.

Scenario I resulted in a steep increase in the number of infected individuals of 27.85 times for Fortaleza (Fig 7(a)), 75.62 times for Rio de Janeiro (Fig 7(c)) and 66.31 times for São Paulo (Fig 7(e)) at the end of the analyzed period, relative to the actual numbers at that same point in time. Initially, at the beginning of the pandemic, the number of infected is tiny. Thus, even using the same set of parameters for $q_0$, the ratio is exactly 1.

In Scenario II we observe a different behavior when compared to Scenario I. Due to mobility restrictions were enforced when the first case occurred, we notice a decrease in the number of infected people notified, up to 0.70 times in Rio de Janeiro, for instance. We observe in Fig 7 (b), 7(d) and 7(f) that, according to our model, the infection would be significantly reduced with a severe mobility restriction since the first sights of the infection. This experiment highlights the importance of joint actions between the government and population to mitigate the adverse effects associated with the COVID-19 pandemic.

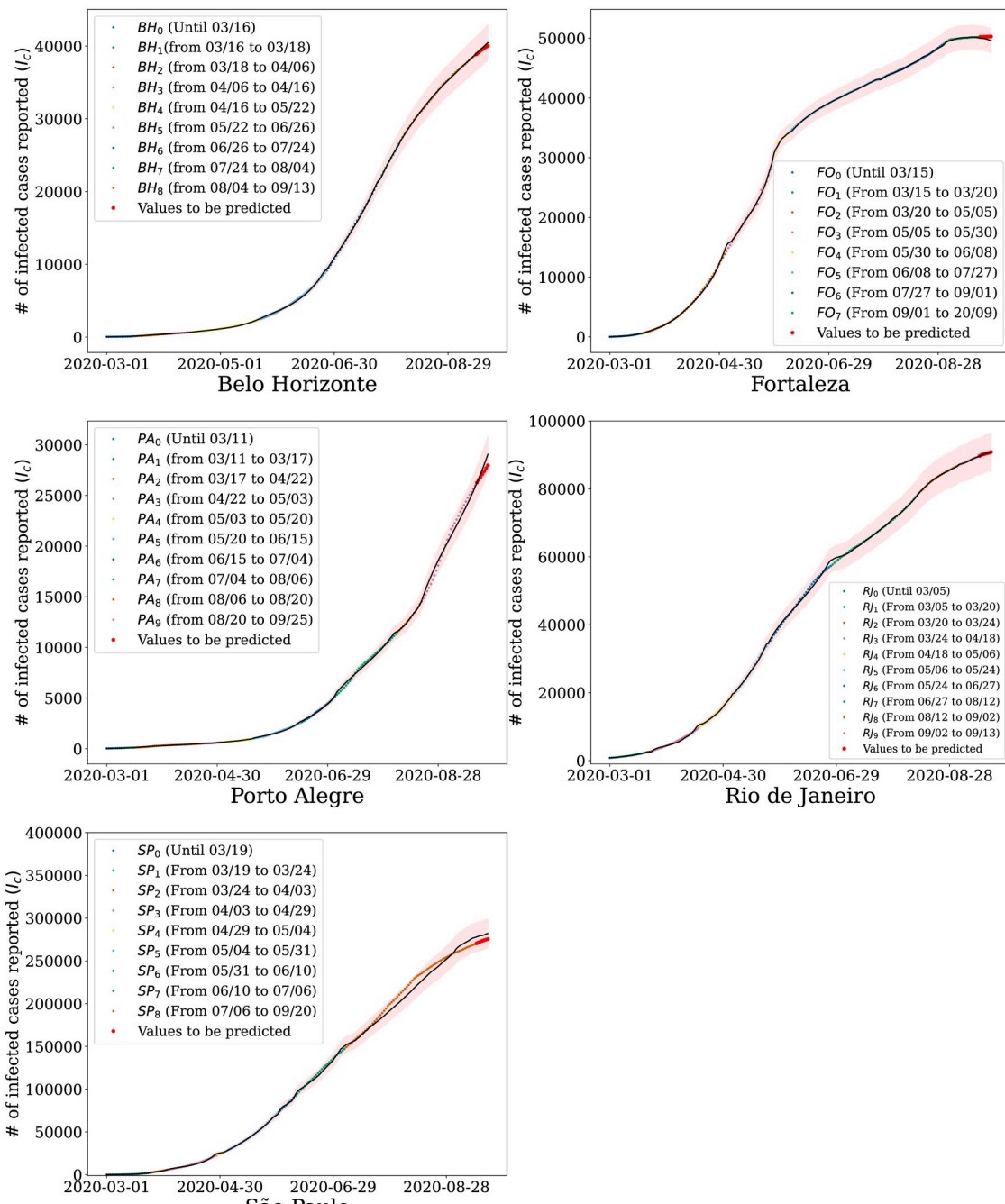

**Fig 5. Simulation results for coarse-grained data.** Number of infected people estimated by our model for the cities of Belo Horizonte, Porto Alegre, São Paulo, Rio de Janeiro, and Fortaleza when analyzed in the context of Waze mobility indexes. The shaded areas represent the 95% confidence region provided by the model; the black line represents the average model prediction and the points the official values released. (a) Belo Horizonte. (b) Porto Alegre. (c) São Paulo. (d) Rio de Janeiro. (e) Fortaleza.

It is worth noting that, although the trade closure may be considered a highly restricted measure, if it is well planned, it has the potential to quicken the end of the pandemic. Further studies to show that such a restriction may lead to fasten the economic recovery are still necessary. It is also worth noting that although the model output indicates that the pandemic would

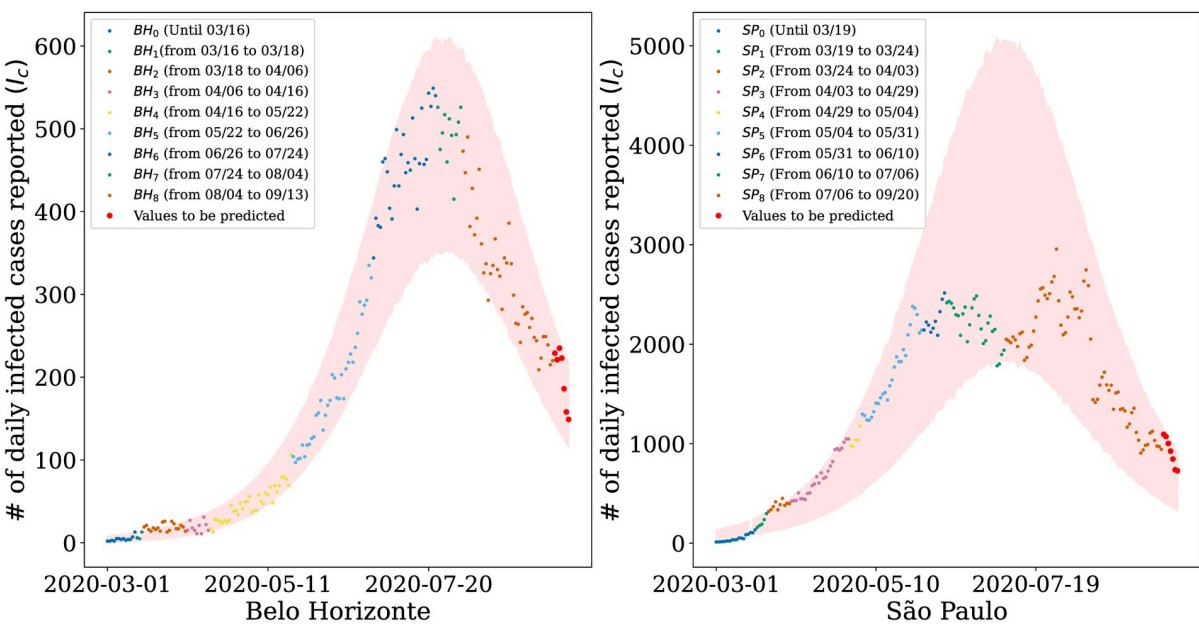

**Fig 6. Simulation results for daily infected cases reported.** Number of infected people estimated by our model for the cities of (a) Belo Horizonte and (b) São Paulo. The shaded areas represent the 95% confidence region provided by the model and the points the official daily values released.

be at its ending by May 2020, even without adopting mobility restriction measures, this would cost millions of deaths and a total collapse of the health system, as the number of infections would rise dramatically. This result does not consider new variants, mutations, or new waves of infection, which could prolong the pandemic.

## Case study II: Fine-grained analysis

We used a dataset of DETRAN-CE's ALPRs logs in Fortaleza to analyze the influence of human mobility in the spread of the COVID-19 in finer granularity. Using this data, we can account for the mobility among neighborhoods since we can reconstruct vehicle routes. Since the dataset was anonymized, we only used the start and endpoints of the vehicle's route to estimate vehicle flows among different regions.

Our dataset comprises annotations informing when a given vehicle passed by the DETRAN-CE's ALPR sensing location. Thus, for the construction of trajectories, we use the checkpoints of the DETRAN-CE's ALPR as follows. We defined the upper limit ($t_s$) when a vehicle passes through a DETRAN-CE's ALPR, but no other DETRAN-CE's ALPR identifies it for an extended period. It probably happened due the vehicle used an alternative route that does not have any DETRAN-CE's ALPR from our dataset. Similarly, the lower limit ($t_i$) is when a vehicle passes two or more DETRAN-CE's ALPR in a short period. This behavior is because the vehicle did not stop anywhere between the DETRAN-CE's ALPRs. Hence, we constructed a trajectory when the measurement between two DETRAN-CE's ALPRs is in the interval $[t_i, t_s]$, indicating that the vehicle stopped at a location between them. We manually investigated the threshold values, and for this work, we use $t_i = 1$ hour and $t_s = 10$ hours.

The first step in carrying out the fine-grained analysis was to group the city's neighborhoods into sub-regions. It increased the model's accuracy compared to using the

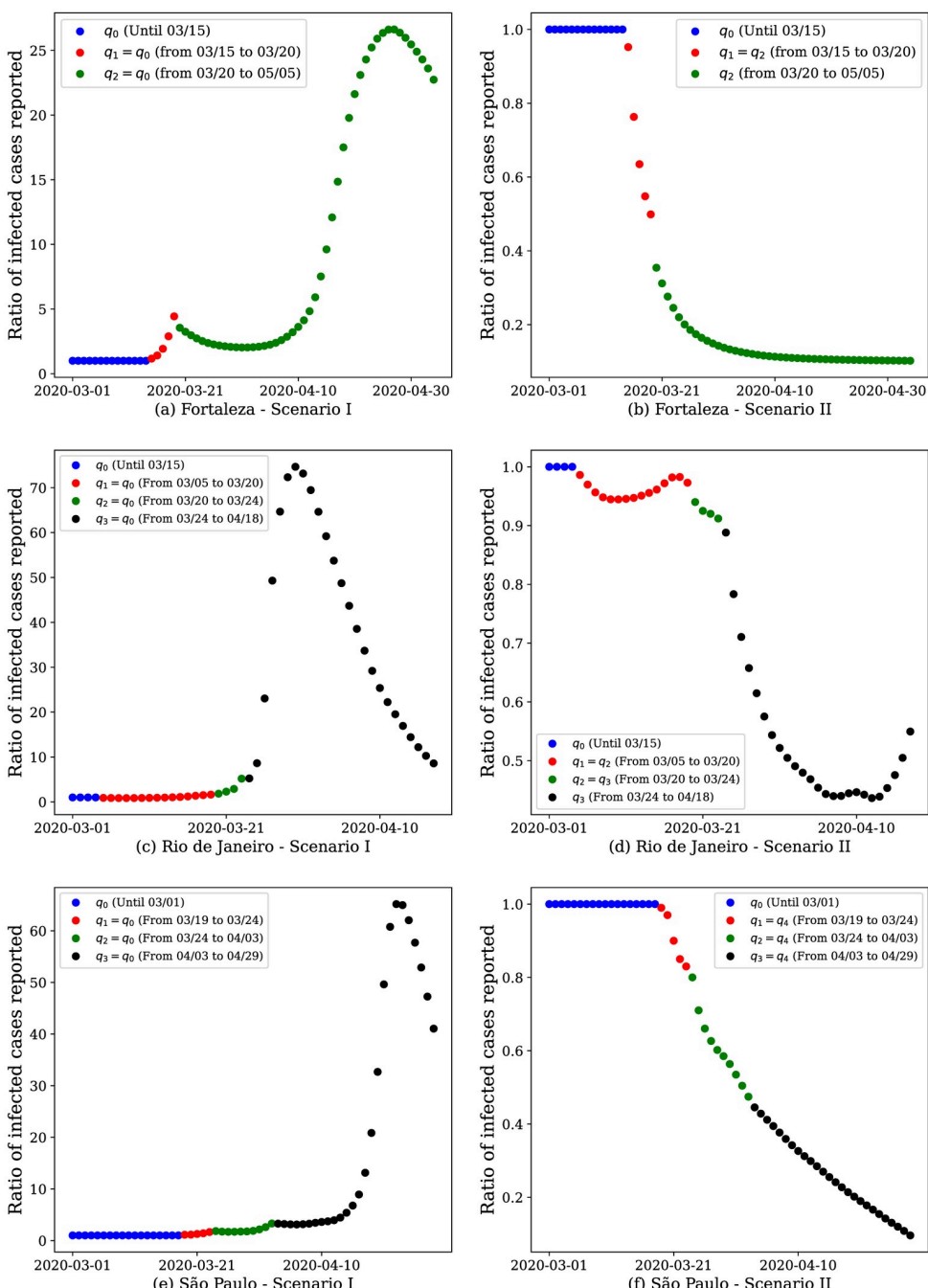

**Fig 7. Results of hypothetical scenarios for coarse-grained data.** The model response when simulating two hypothetical scenarios: I) the population and neither the government prioritized measures to restrict mobility and II) after the first case notified in the city, the government decreed the closure of trade. Here we analyze the ratio between the number of infected individuals in the analyzed scenarios and the actual number of infected individuals to observe the curve trend over time.

neighborhoods alone since the number of variables to be estimated is reduced, making us better capable of capturing the impact of contagious transitions and outbreaks between regions.

Evaluating the contagion curve of COVID-19 in Brazil, we observed that there is a relation between the mortality rate in a public Intensive Care Unit (ICU) and private hospitals [35].

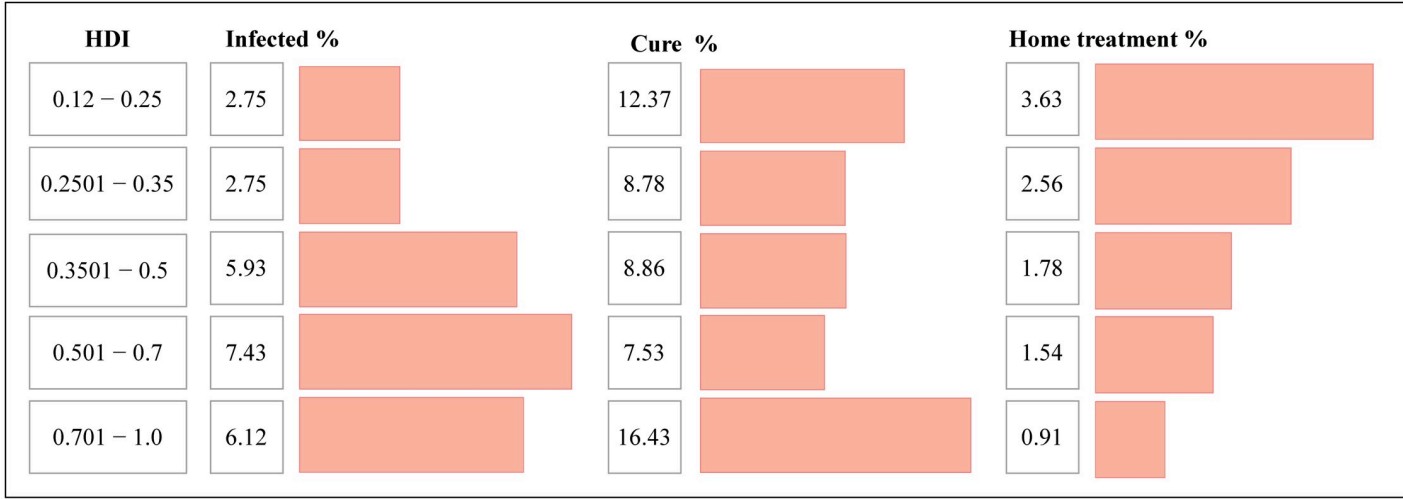

**Fig 8. Social analysis of the pandemic spread to the city of Fortaleza.** Results of the social analysis of the spread of the virus under the context of HDI in the neighborhoods of the city of Fortaleza. As we can see, we found that the higher the HDI, the higher the percentage of infected people in the region, as well as the higher recovery, which we believe is directly associated with the greater number of patients with access to quality private hospital treatment. On the other hand, in regions with the lowest HDI rates, we observed the highest percentage of treatments performed independently, which can be explained by the lack of access to public service.

Analyzing the HDI of neighborhoods in Fortaleza, we note some characteristics about the virus spread. As we can see in Fig 8, a higher rate of infected people occurred in regions with the highest HDIs, confirming the worldwide trend. However, these regions with the highest HDIs also have a lower index of home treatment, indicating that such patients had more access to the emergency units and ICUs than residents of less favored regions. With more access to specialized hospital treatment, we also see a higher recovery rate among these individuals.

Since our objective here is to aggregate similar regions, we assume a relationship between the HDI and the recovery from the disease. The social clusters applied in this work were built based on two local characteristics: geographic connectivity, and the HDI. We group 52 neighborhoods present in DETRAN-CE's dataset into 16 regions formed by adjacent neighborhoods that have similar HDI and mobility indexes. Table 2 presents the aggregated regions and their respective neighborhoods.

Note that, to calibrate the values of regions with final granularity, we use Opendata SUS. Therefore, we consider the ratio $I_{N_i}(t)/I_{N_j}(t)$ for calibration/sensitivity of the $q_i(t)$ element, where $I_{N_j}(t)$ is the number of infected individuals notified from the region with the lowest number of cases for a day $t$. In other words, our model estimates the parameters so that its prediction keeps the ratio adopted in the Coronavirus Panel data. We adopt the relative value since the Opendata SUS dataset has a small sample of data. Furthermore, we consider for calibration only the days where $I_{N_j}(t) > 10$.

Due to the characteristics of the analyzed disease, we expected that a change in human mobility dynamics impacts the infection rate with a specific time lag. In other words, changes in mobility do not immediately impact the infection rate. As seen in Fig 10, as the reproduction number decreases, the normalized value of the daily trajectory frequency presents an increasing behavior. We can verify this relationship through cross-correlation tests, where we find a maximum correlation when $lag = -1$, as can be seen in Fig 9.

This result means that when we backward the data corresponding to the vehicle flow in one day, we can see a negative correlation of 44.9% between mobility and the COVID-19

**Table 2. Descriptions of the regions and their respective neighborhoods used in the proposed model.**

| Regions | Neighborhoods |
| --- | --- |
| RE1 | Parque dois irmãos, Arvoredo pequeno Mondubin, Parque novo Mondubin, Castelão and Passaré |
| RE2 | São João do Tauape, Sapiranga, Edson Queiroz and Manoel Dias Branco |
| RE3 | Antônio Bezerra, Padre Andrade, Jardim Guanabara and Alvaro Weyne |
| RE4 | Barra do Ceará, Cristo Redentor and Jardim Iracema |
| RE5 | João XXIII and Bom Sucesso |
| RE6 | Parangaba, Itaperi and Maraponga |
| RE7 | Serrinha, Alto da balança and Cajazeiras |
| RE8 | Bom futuro, Jardim Américo and Rodolfo Teofilo |
| RE9 | Benfica, Damas, Centro and Parquelândia |
| RE10 | José Bonifácio, joaquim Tavora, Fátima and Parreão |
| RE11 | Praia de Iracema, Dionisio Torres, Aldeota and Meireles |
| RE12 | Farias Brito, Monte Castelo, Jacarecanga and Alagadiço novo |
| RE13 | Barroso |
| RE14 | Engenheiro Luciano Cavalcante |
| RE15 | Moura Brasil |
| RE16 | Vicente Pizon |

transmission (calculated with the data adjusted by the time-variation delay). This behavior is likely to happen because when the infection rate decreases, individuals tend to feel more confident about getting out of social isolation and consequently increasing their mobility. As can be seen in Fig 10, as the virus transmission rate decreases, we observe an increase in the vehicle flow, indicating greater mobility of the population.

By performing cross-correlation tests between the aggregate vehicle flow frequency from DETRAN-CE's data and the indices in the different categories of places present in Google COVID-19 Community Mobility Reports, we can better understand what type of mobility behavior these data capture. In Fig 10, we see correlated patterns between the data from DETRAN-CE and some other mobility categories. The highest correlation indexes presented were obtained with $lag = 0$ in recreational areas (with 78.9%), parks (with 74.9%) and grocery stores (with 61.1%). Thus, we observed that the greater the vehicle flow captured by DETRAN-CE's ALPR, the greater the mobility in outdoor leisure areas and supermarkets, indicating a greater preference of the population for getting around with automotive vehicles to/from such environments.

Given the type of movement captured by our data, our next step was to investigate the causal direction between two variables analyzed using the Granger causality test. We were able to verify that the past of the vehicle flow, delayed 1 day, helps to predict the present value of $R(t)$ of COVID-19, indicating a causal sense among them. So, the flow of vehicles "Granger causes" the reproduction number $R(t)$ delayed by 1 day, i.e., even after the temporal re-alignment of the data, we have to correct DETRAN-CE's vehicle flow data in one unit.

To evaluate the model, we investigate the usefulness of our methodology for bounding the number of cases in the near future. Specifically, we identified three critical events by evaluating our model on 16 regions in the city of Fortaleza from 03/15/2020 to 05/04/2020:

- (03/15/2020)—Occurrence of the first notified case of COVID-19.

- (03/20/2020)—Government enforces the trade closure.

- (05/05/2020)—Beginning of a lockdown.

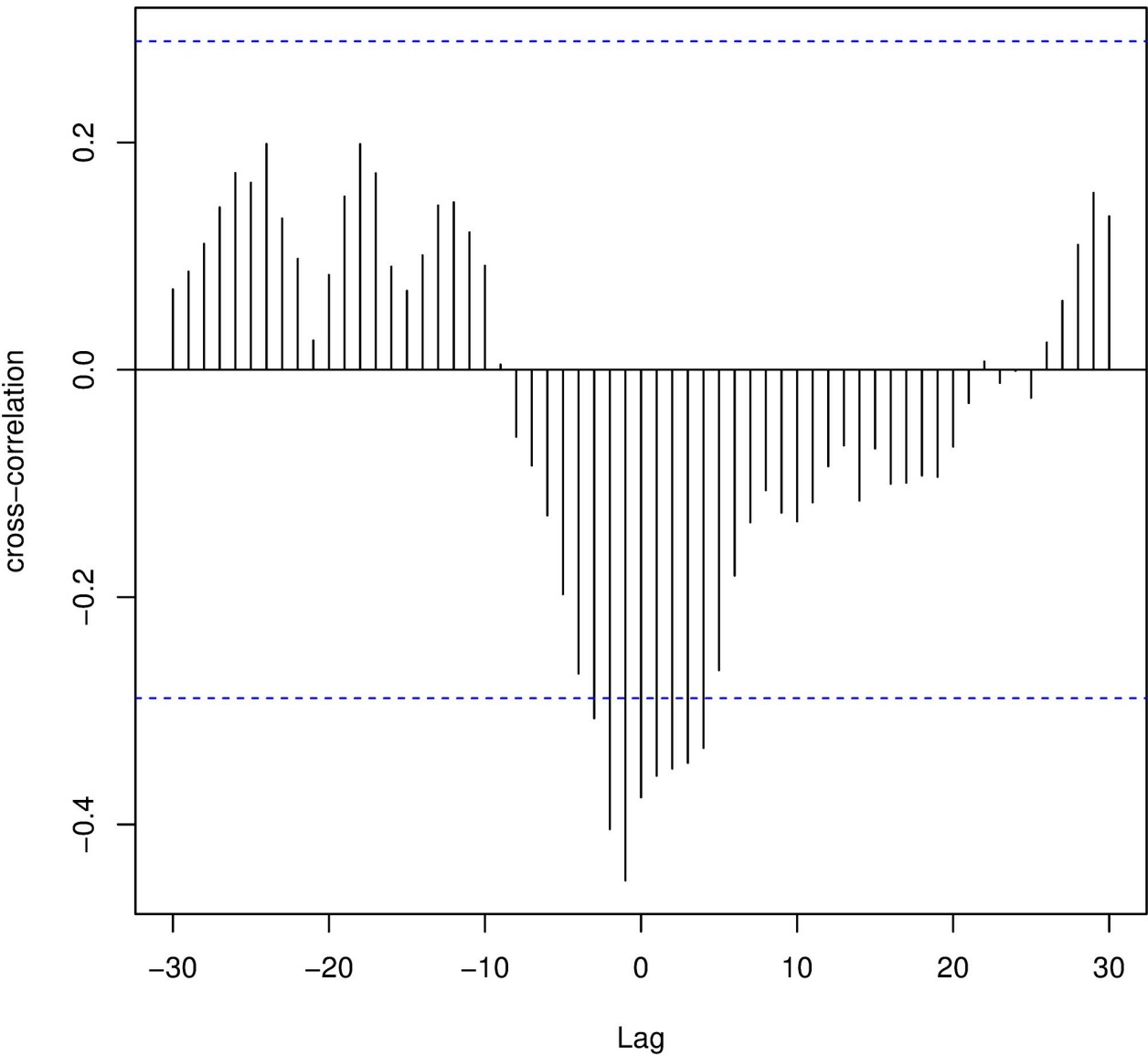

**Fig 9. Cross-correlation between mobility data and disease spread.** Representation of the cross-correlation between $R(t)$ is the cars' flow obtained by DETRAN-CE. We observe that, the correlation is maximum (in module) for *lag* = −1.

Considering these events and the availability of trajectory data for Fortaleza up to 05/04/2020, we define three periods for investigation: before the pandemic, between the first notified case and the trade closure, and between the trade closure and the beginning of a lockdown. To avoid the potential effects of the COVID-19 news on mobility during the weeks preceding the first notified case, we set the first period to be the interval between 01/16/2020 and 02/29/2020. Thus, the flow matrix $C_{ij}$ is obtained through the frequency of daily trajectories between the analyzed regions.

For each period, we estimate the corresponding mobility parameter $q$. More specifically, we set:

- $q = q_0$: Mobility indicator that represents the typical behavior of the population of Fortaleza, hence $q_0 = 1$.

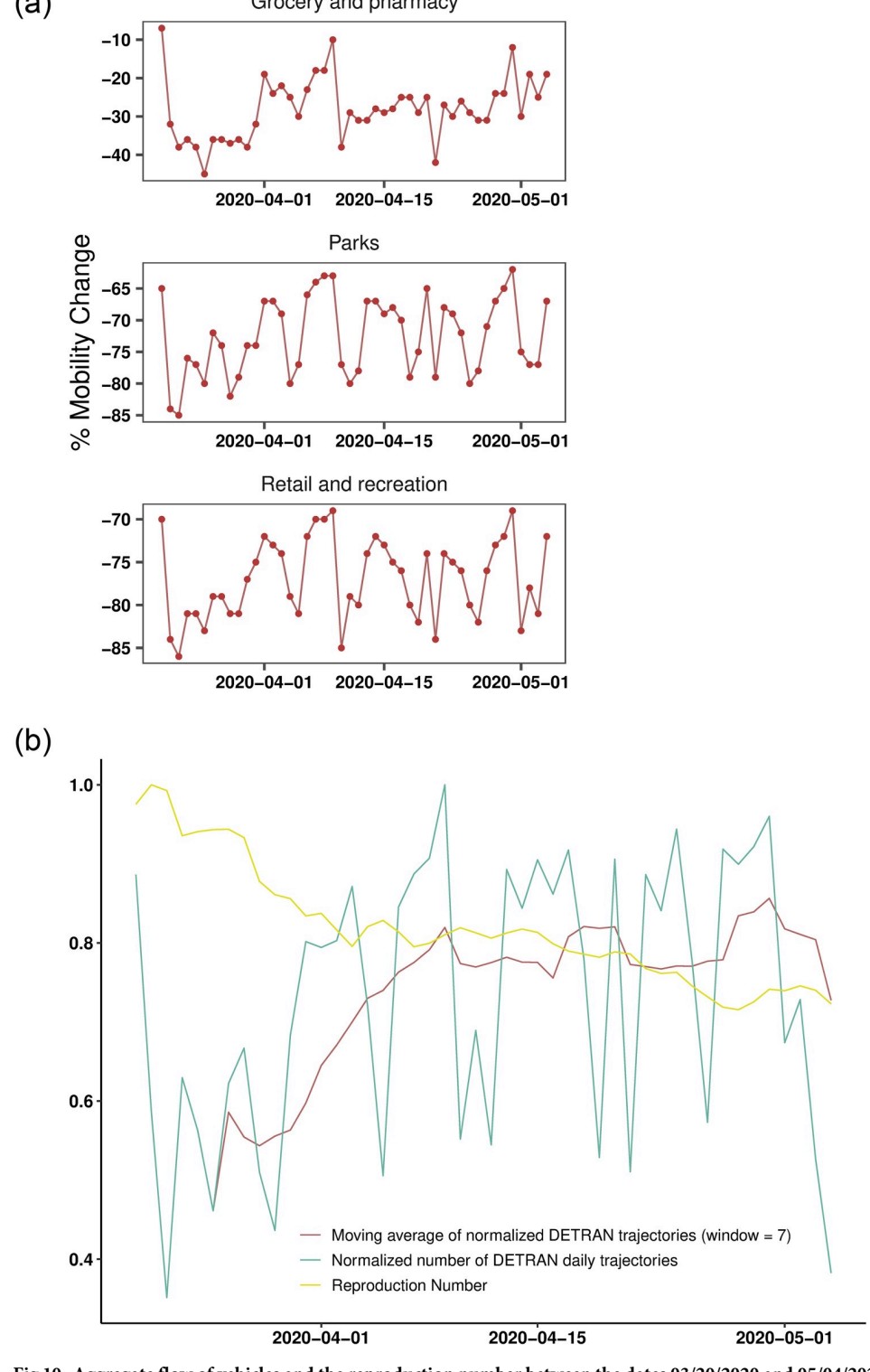

**Fig 10. Aggregate flow of vehicles and the reproduction number between the dates 03/20/2020 and 05/04/2020.**
Plots in the left show mobility indexes extracted from the Google report, and plots in the right depict the *R(t)* estimated
from DETRAN-CE's data. We can see similarities between trends in DETRAN-CE's data and the mobility indexes
extracted from the Google report. The highest cross-correlation results are in places labeled as retail and recreation,
grocery and pharmacy, and parks.

- $q = q_1$: Mobility indicator that represents the period between the first case notified in the city of Fortaleza and the closing of the trade.

- $q = q_2$: The mobility indicator represents the period between the closing of the trade and the lockdown.

To investigate the adherence of our model to the data, we again removed the last 7 days present in our dataset and analyzed the model's response, as depicted in Fig 11.

Although SENUR model without mobility can satisfactorily capture the pandemic behavior, it cannot answer questions regarding the impact of mobility. For instance, we cannot quantify how the government measures of mobility restriction impact on the infection rate. Therefore, we propose in this work an extension of this model to investigate human mobility's influence on the spread of the COVID-19 pandemic. We can observe in Fig 11(a) that the confidence interval is wider than the one presented in Fig 11(b). It is an evidence that our model captures part of the variability of the model by estimating the mobility. Therefore, although both models can be satisfactorily used, our model is more expressive in terms of mobility and it is more accurate for the analyzed data.

As we are using a regional approach, we can stratify the information for each region individually. In this way, we infer how the pandemic affects the city of Fortaleza with finer granularity and, by doing so, we can analyze how each of these regions influences the pandemic through its $R(t)$. We see in Fig 12 that the regions have different behaviors and, thus, allows us to optimize actions individually to contain the pandemic, such as proposing more severe restrictions in high-risk regions or those more geographically connected. For instance, we observe that around 04/06/2020, region RE1 presents an increase of $R(t)$ steeper than region RE11. Other regions present similar behavior.

We can see the peak in the graphs in Fig 12. After an investigation, we observed that Fortaleza changed the perception of Covid-19 after the death of a 3-month-old baby (on 03/04)

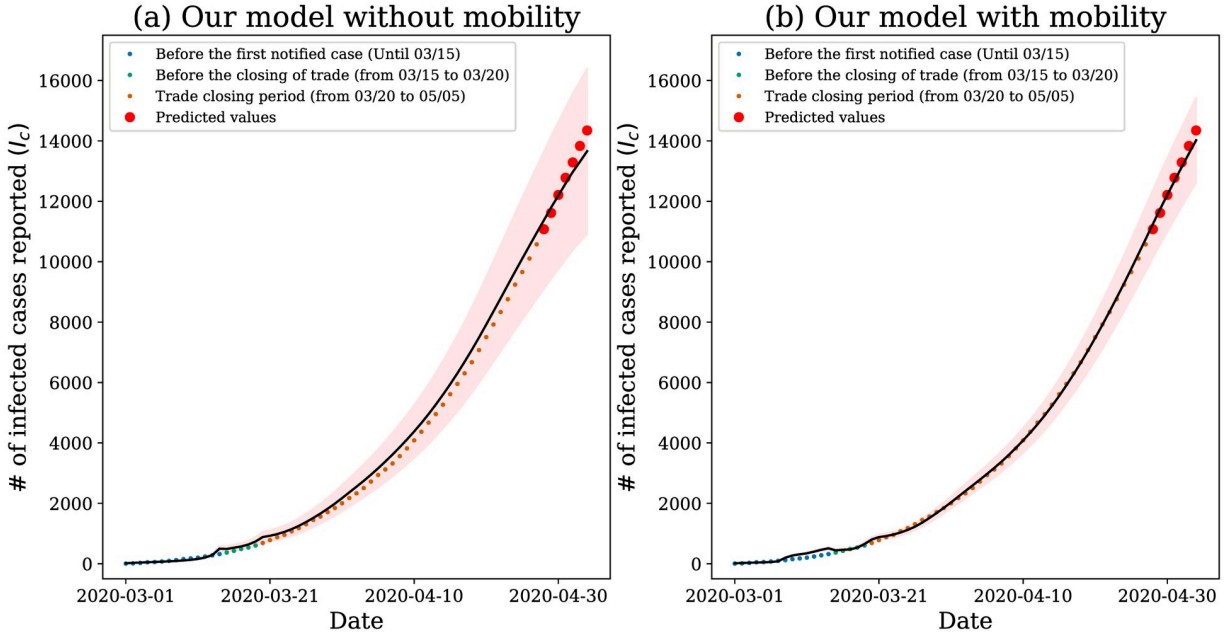

**Fig 11. Model results.** Our model applied to data from the city of Fortaleza. The shaded areas represent the 95% confidence region provided by the model; the black line represents the average model prediction and the points the official values released.

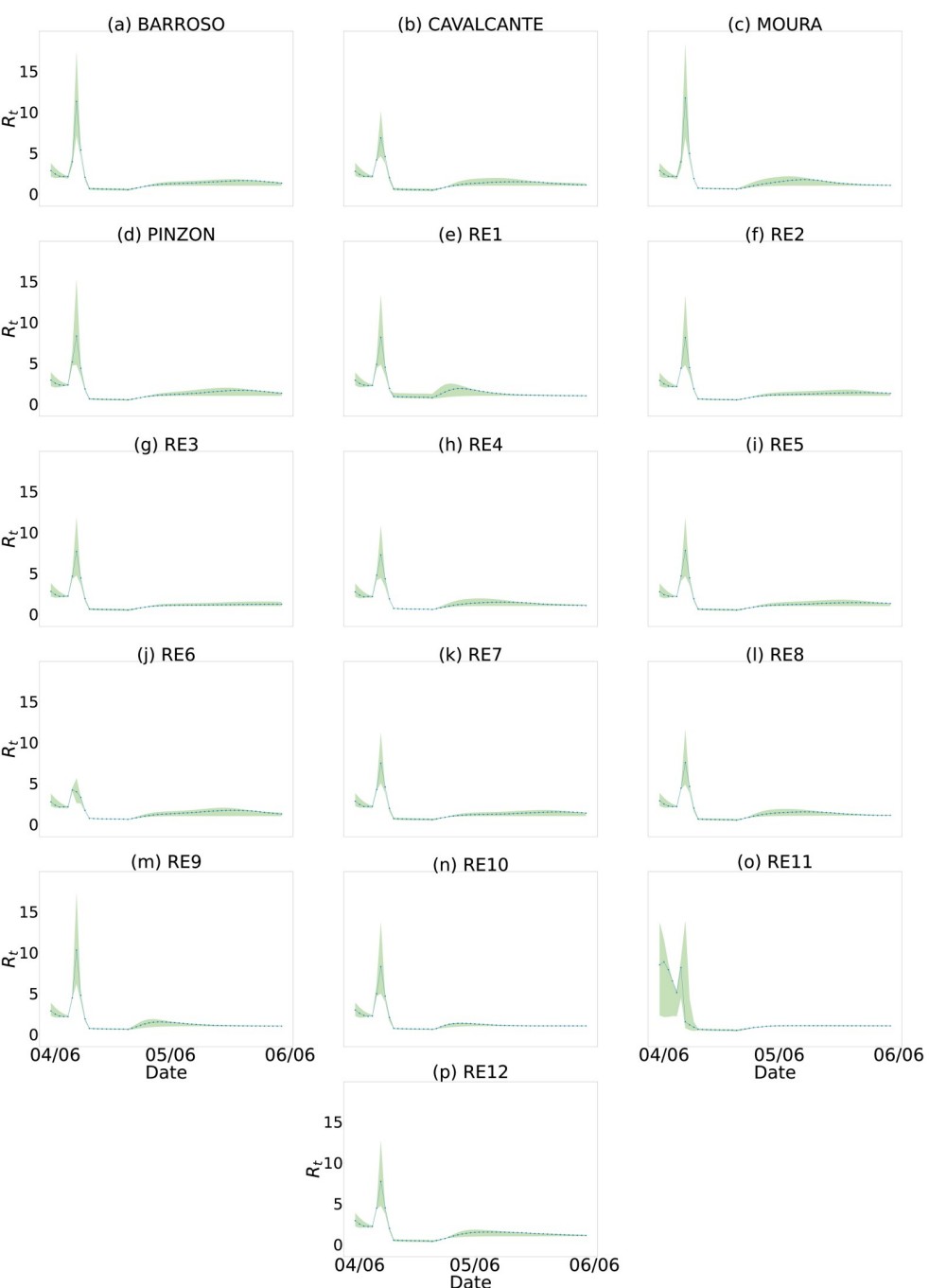

**Fig 12. $R(t)$ estimate for all 16 regions of the city of Fortaleza.** We use the number of infected cases estimated by our model for each of the regions. (a) BARROSO. (b) CAVALCANTE. (c) MOURA. (d) PINZON. (e) RE1. (f) RE2. (g) RE3. (h) RE4. (i) RE5. (J) RE6. (k) RE7. (l) RE8. (m) RE9. (n) RE10. (o) RE11. (p)RE12.

[36]. On 04/04, the state government released a statement [37] about this fact and reinforced the pandemic's seriousness. The great commotion is likely to be related to the steep drop in the value of the observed $R(t)$.

Therefore, governments can use this kind of knowledge to design public policies to contain the pandemic. This kind of analysis is possible by using our model on fine-grained data.

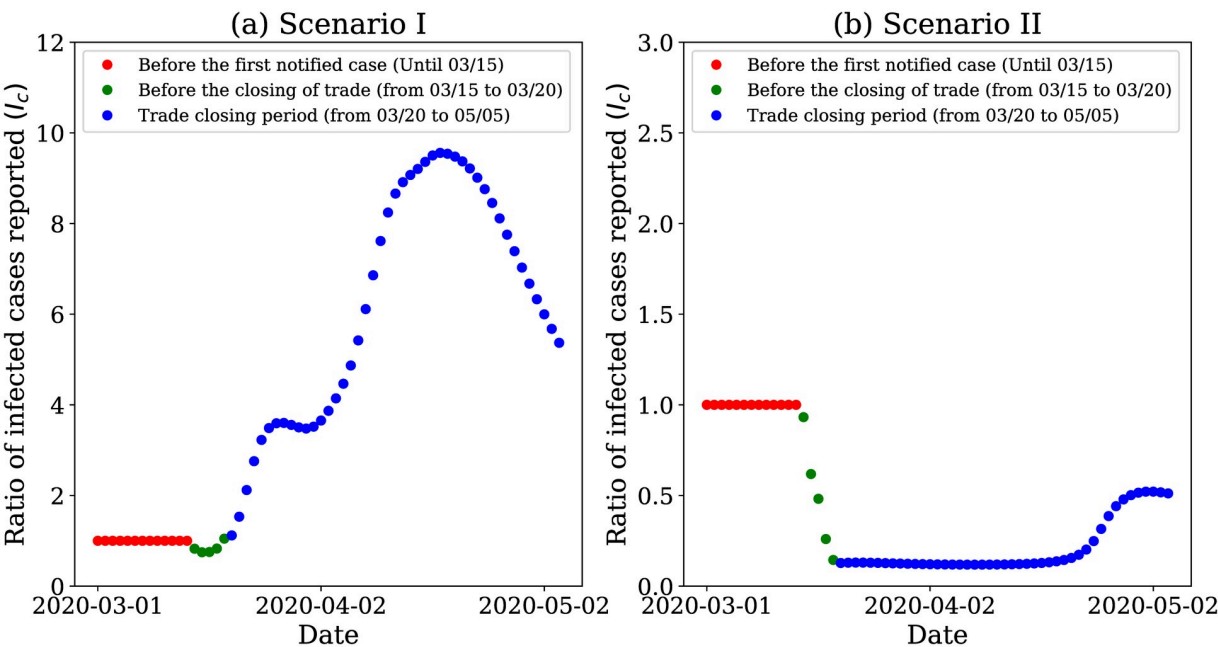

**Fig 13. Results of hypothetical scenarios for fine-grained data.** Model's response when simulating hypothetical scenarios in Fortaleza when we apply fine-grain spatial mobility data.

To analyze the mobility impact, we again investigate the pandemic behavior for two hypothetical scenarios of increasingly restrictive interventions, as can see in section **Case study I: Coarse-grained analysis**. Scenario I increased the number of infected individuals concerning the actual number of 9.15 times at the end of the analyzed period—Fig 13(a). On the other hand, in scenario II, we observed the reduction in the number of infected individuals to 0.52 times of the actual infected number—Fig 13(b). In this hypothetical scenario, the risks would be more negligible, and the curve would flatten. We also observe that with a more restrictive measure, since 03/11/2020, the number of infected cases would decrease; hence, we can say that the pandemic would be under control.

Governments may use the analysis shown in Fig 12 to monitor the behavior of the pandemic and relax the restriction measures. Every time the $R(t)$ of any region starts to increase, the government must rapidly decree restrictive measures to prevent the pandemic from increasing.

## Sensitivity analysis

In this work, we estimate a mobility factor $q_t$ from our model to quantify the mobility dynamics of the COVID-19 pandemic in some Brazilian cities. In order to assess the sensitivity of our proposal, we compared our mobility quantifier with some real mobility data, as shown in Fig 14.

Initially, we compared $q_t$ with the flow of vehicles in the city of Fortaleza. Then, we compared the estimated values of $q_t$ in section **Case study II: Fine-grained analysis**, with the periods analyzed at the average normalized number of DETRAN-CE daily trajectories. We observed the trend between these two components, where in the first period $q_0 = 1$, we have the value of 0.8651 found in the normalized mean series of the vehicle flow, which is the highest value among the analyzed periods. As the mobility index value decreases, as we can see in

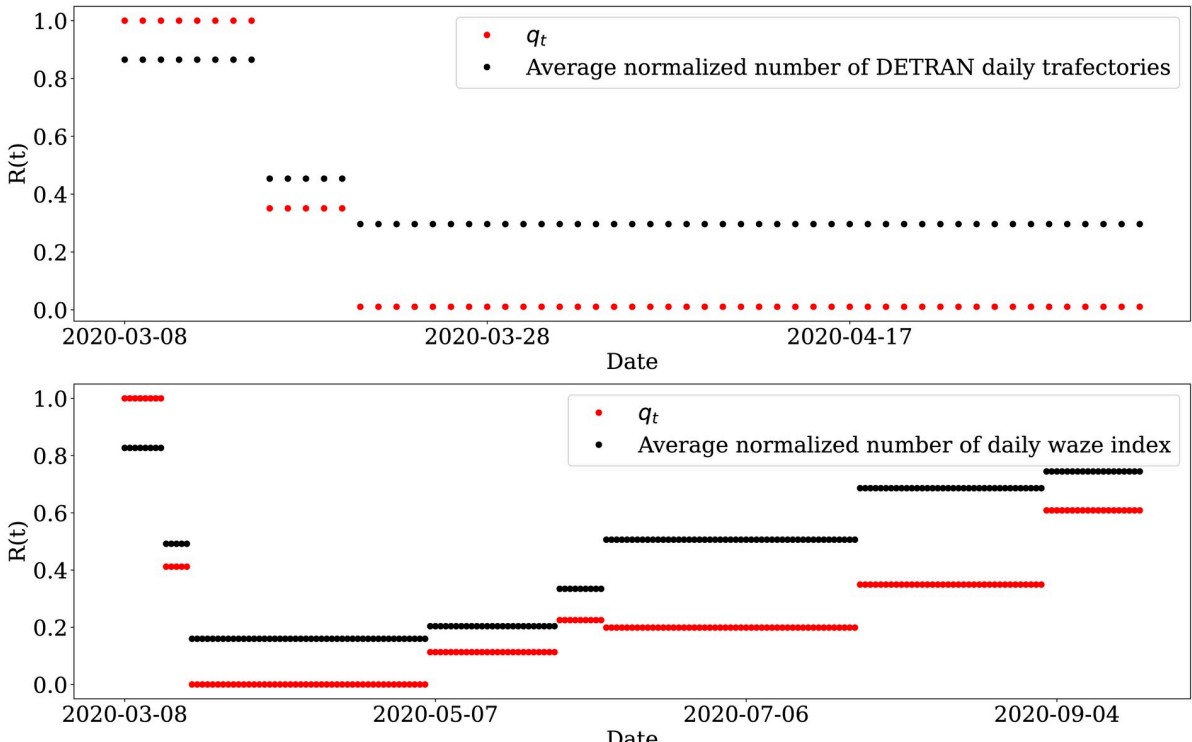

**Fig 14. Sensitivity analysis.** Comparison between our mobility quantifier with some real mobility data used in our work.

$q_1 = 0.351$ and $q_2 = 0.0106$, we find that the average normalized of the vehicle flow also decreases to 0.4537 and 0.2966, respectively.

We also analyzed the case with coarse granularity. In this case, we compared the value of $q_t$ with the normalized mean of the Waze mobility index. In this case, we observe the same trend, i.e., excluding the period 06/08 to 07/27 ($q_5$), as the value of $q_t$ increases/decreases, the value of the normalized mobility index of the Waze increases/decreases respectively.

Furthermore, it is worth noting that for the period used in the two figures, the values of $q_t$ are corresponding; that is, for the case of fine granularity, we find $q_1 = 0.371$ and $q_2 = 0.0106$ while for the case of coarse granularity we have $q_1 = 0.417$ and $q_2 = 0.009$, indicating that our technique estimates similar values for the same period, even using different modeling and data sources.

It is worth noting that the $q_t$ is a parameter estimated by our model. In our analysis, we estimated $q_t$ every time a mobility restriction/release was announced, however, $q_t$ can be estimated periodically in the case it is more adequate. For instance, $q_t$ can be estimated daily, weekly or biweekly, similarly to the usual $R(t)$ estimation.

## Conclusion

This work proposes investigating the benefits of analyzing mobility under different granularities in disseminating COVID-19 in the Brazilian territory. We also observed the importance of mobility and isolation measures in determining the infection curve through the simulation of hypothetical data and correlation and causality tests. Without restrictions, social isolation based on the fear of contagion would only delay the peak of the contamination curve. On the other hand, when the mobility restrictions were introduced shortly after the confirmation of

the first case of contagion, in addition to delaying the occurrence of the peak, the resulting curve was significantly flatter than in the other scenarios.

Because of this, we found that the finer-grained mobility information can significantly support the public policy decision-making process, making governments take decisions that prevent the collapse of the health system and, consequently, preserve people's health. Besides, governments can have more information to make decisions to prevent economic harm. We are aware that fine-grained data are difficult to access; however, we also presented an alternative study based on public data provided by large Internet companies.

As future work, we leave the investigation of different vaccination scenarios in the current context of the Brazilian territory. In this scenario, it would be necessary to stratify the model by different age groups, making it possible to study the priority groups for vaccination and model the estimates of immunization effectiveness and time.

## Acknowledgments

The authors sincerely thank Cristopher G. S. Freitas and Alejandro C. Frery for their helpful comments on the present research.

## Author Contributions

**Conceptualization:** Eduarda T. C. Chagas, Pedro H. Barros, Isadora Cardoso-Pereira, Pablo Ximenes, Flávio Figueiredo, Fabricio Murai, Ana Paula Couto da Silva, Jussara M. Almeida, Antonio A. F. Loureiro, Heitor S. Ramos.

**Data curation:** Igor V. Ponte, Pablo Ximenes, Antonio A. F. Loureiro, Heitor S. Ramos.

**Formal analysis:** Eduarda T. C. Chagas, Pedro H. Barros, Isadora Cardoso-Pereira, Flávio Figueiredo, Fabricio Murai, Ana Paula Couto da Silva, Jussara M. Almeida, Heitor S. Ramos.

**Funding acquisition:** Heitor S. Ramos.

**Investigation:** Eduarda T. C. Chagas, Pedro H. Barros, Isadora Cardoso-Pereira.

**Methodology:** Eduarda T. C. Chagas, Pedro H. Barros, Isadora Cardoso-Pereira, Flávio Figueiredo, Fabricio Murai, Ana Paula Couto da Silva, Jussara M. Almeida, Antonio A. F. Loureiro, Heitor S. Ramos.

**Project administration:** Fabricio Murai, Ana Paula Couto da Silva, Jussara M. Almeida, Heitor S. Ramos.

**Resources:** Heitor S. Ramos.

**Software:** Eduarda T. C. Chagas, Pedro H. Barros.

**Supervision:** Flávio Figueiredo, Fabricio Murai, Ana Paula Couto da Silva, Jussara M. Almeida, Antonio A. F. Loureiro, Heitor S. Ramos.

**Validation:** Eduarda T. C. Chagas, Pedro H. Barros, Flávio Figueiredo, Fabricio Murai, Ana Paula Couto da Silva, Jussara M. Almeida, Antonio A. F. Loureiro, Heitor S. Ramos.

**Visualization:** Flávio Figueiredo, Fabricio Murai, Ana Paula Couto da Silva, Jussara M. Almeida, Heitor S. Ramos.

**Writing – original draft:** Eduarda T. C. Chagas, Pedro H. Barros, Isadora Cardoso-Pereira.

**Writing – review & editing:** Eduarda T. C. Chagas, Pedro H. Barros, Isadora Cardoso-Pereira, Igor V. Ponte, Pablo Ximenes, Flávio Figueiredo, Fabricio Murai, Ana Paula Couto da Silva, Jussara M. Almeida, Antonio A. F. Loureiro, Heitor S. Ramos.

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
