## [Decision Letter · Decision Letter 0]

8 Jun 2021

PONE-D-21-13982

Effects of  population mobility on the COVID-19 spread in Brazil

PLOS ONE

Dear Dr. Ramos,

Thank you for submitting your manuscript to PLOS ONE. After careful consideration, we feel that it has merit but does not fully meet PLOS ONE’s publication criteria as it currently stands. Therefore, we invite you to submit a revised version of the manuscript that addresses all the points raised in the two excellent reviews.

We look forward to receiving your revised manuscript.

Kind regards,

Sergio Gómez

Academic Editor

PLOS ONE

Journal Requirements:

Reviewers' comments:

Reviewer's Responses to Questions

**Comments to the Author**

1. Is the manuscript technically sound, and do the data support the conclusions?

Reviewer #1: Partly

Reviewer #2: Partly

2. Has the statistical analysis been performed appropriately and rigorously? 

Reviewer #1: Yes

Reviewer #2: I Don't Know

3. Have the authors made all data underlying the findings in their manuscript fully available?

Reviewer #1: No

Reviewer #2: No

4. Is the manuscript presented in an intelligible fashion and written in standard English?

Reviewer #1: Yes

Reviewer #2: Yes

5. Review Comments to the Author

Reviewer #1: The authors present a work on the effect of human mobility during the COVID-19 epidemic in Brazil. The results presented are reasonable and in agreement with similar studies in the present literature, the methodology is standard and quite straightforward, the data collected are interesting overall.

As regards the content and the results achieved, I think the paper has the potential for being published in Plos One, however it most definitely needs a major revision in the presentation style, which is currently poor and not adequate for a scientific publication in general. The paper overall seems to have been written in a rush, the plots in particular are extremely unclear

and poorly presented. I invite the authors to revise the main text, and to rethink and reorganise all current figures.

Here my concerns:

1) In general, simply put, the plots are barely readable. First, the quality of the images is very poor: even when downloaded the figures appear heavily pixelated with very low resolution. I suggest the authors to use vector graphics systematically. Secondly, I strongly suggest the authors to arrange multiple plots regarding the same quantity in large single panels, instead of providing a 10+ pages with one plot for each single page. For instance, the 16 plots for the Rt on different regions could be easily arranged in a single panel with 16 plots. Also, the figures appear in order 10-11-12-1-2-3-..., which is even more confusing.

Now, as regards the single figures:

- Figure1a: There are no labels in the axes. Also, it is not specified what does the dotted vertical line represent, neither in the plot nor in the text. Mean value? Median? Please, specify.

- Figure1b: Same as figure 1a.

- Figure 1c: No need to use a Y scale this large.

- Figure 2: Compartments I_Ni and I_Vi do not appear in the equations in the text. To this matter see also point 4.

- Figure5(all): The legend overlaps the points. Confidence intervals (when they appear), look wrong. The points always sit at the extreme values of the confidence intervals, which does not look correct. My best guess is that one side of the interval is missing, please check the correctness of those confidence intervals.

- Figure6(all): Why are the curves discontinuous? Also, the legend in each plot seem to be wrong, for instance in figure 6a q2 refers to both the red and the green curve. This ambiguity is present in all plots of figure 6 and gets even worse, for instance in figure 6c all curves are labeled as q0. Please, clarify or correct.

- Figure9a: Missing label on y-axis.

- Figure 9b: What are the unit of the DETRAN line? What does it represent? How is computed? Why is it compared against Rt?

- Figure 10: I guess that "values to be predicted" means "predicted values". Also, the curve looks like a simple exponential or a power law. How does the model prediction would compare with a simple exponential fit for instance?

- Fig11(all): Why Rt stops at 5? Where does it go?

- Fig 12(all): Again, I don't see we the lines are discontinuous. Also, what are the units on why axes? Seems very odd that the number of case reported peaks at 10 (Fig.12a).

2) Line 27: Avoid using the phrasing "critical transition" if it cannot be formally justified. Critical transitions are well defined in statistical physics and need to satisfy particular properties which I don't think apply to the case study.

3) Eq.(1-5) is a technically a system of Ordinary Differential Equations (ODEs), not Partial Differential Equations (PDEs). PDE

are usually invoked when dealing with multivariable functions. On the other hand, in this case each compartment is only a function of time.

4) Infected sub-compartments are presented as I_Ni and I_Ui, but in the equations appear I_Ci ans I_Si, which are not defined. What are these compartments? How they relate to the previous ones?

5) Why should d_N and d_U be different?

6) In the expression for lambda_i a mu appears, not defined. Please, clarify what mu is.

7) What is an internal mobility index?

8) What are the possible values for q_i in general?

9) Line 336: 2785% with respect to? Same apply for every other percentage. Please clarify the meaning of every percentage.

10) The English overall can be largely improved.

Reviewer #2: PLOS ONE: Effects of population mobility on the COVID-19 spread in Brazil (2021), v1

# Summary

The manuscript investigates the SARS-CoV-2 spread and its relationship with the policies adopted in Brazil using mobility data, aiming to show how this data can be used to understand the virus propagation. A SENUR compartmental model is used to accommodate such data with parameters to study hypothetical scenarios of mobility restrictions. The authors used aggregated (Waze) and fine-grained (DETRAN-CE) data.

Although I think this data can be useful, I am not convinced that the authors used it in a way to allow to distinguish the main results from a more basic model, without mobility data. As many things were not clear, I suggest a heavy revision of the manuscript, according to the points I make below, together with the fact that I found that the ideas were put in a very confusing way. An even more systematic revision should be done on the figures, which do not follow a style pattern, either without labels or using different names for the same metrics.

One of the PLOS ONE's criteria for publication is that *"Experiments, statistics, and other analyses are performed to a high technical standard and are described in sufficient detail."* I do not see that in the present version of the manuscript. No sensibility analysis was performed to show the present correlations between the parameters defined for the model, which does not allow drawing the conclusions stated by the authors.

Finally, my opinion is that the manuscript should be rewritten with a better organization and statistical analysis to support the conclusions.

# General comments

In the abstract, it is stated that "This work is the first to shed light on the pandemic situation on the

Brazilian territory using both aggregated (...) and fine-grained (...) mobility data (...)": It is not true. Here I cite some works that investigated this issue in Brazil using these kinds of data:

- "Assessing the potential impact of COVID-19 in Brazil: mobility, morbidity and the burden on the health care system." medRxiv 2020.03.19.20039131 (2020)

- "Evolution and epidemic spread of SARS-CoV-2 in Brazil." Science 369.6508 (2020): 1255-1260.

- "Modeling future spread of infections via mobile geolocation data and population dynamics. An application to COVID-19 in Brazil." PLOS ONE 15.7 (2020): e0235732.

- "Outbreak diversity in epidemic waves propagating through distinct geographical scales." Physical Review Research 2(4) (2020): 043306.

- "Spatiotemporal pattern of COVID-19 spread in Brazil." Science 372.6544 (2021): 821-826.

None of these works are on the reference list.

The mobility restrictions seem to be a change in the infection rate, as a "social distancing" measure or other NPIs. If the authors had mobility data as a function of time (that was not clear to me), such as the data available in

- "Heterogeneous impact of a lockdown on inter-municipality mobility." Physical Review Research 3.1 (2021): 013032.

- "COVID-19 lockdown induces disease-mitigating structural changes in mobility networks." Proceedings of the National Academy of Sciences 117.52 (2020): 32883-32890.

it would be interesting. However, the model is based on a parameter $q_t$ that is calibrated and no sensitivity or calibration analysis was presented to show the possible correlation between different parameters. I am not convinced that the mobility data used were important to draw conclusions.

No discussion was made about each city's results, of how the measures adopted there were efficient or not to mitigate the spread. There is only a discussion about how the numbers would change if different strategies were adopted.

Figures: Each figure uses a different style and notation. Not even the legends of the figures were adapted to fit the space, and in some cases the legend hides the curves or points.

# Specific points:

- Abstract) What the authors mean by "data from public sources"? The data is not available publicly and this phrase sounds like it is.

- p.1) "The negative effects of the COVID-19 pandemic in Brazil may be related to the lack of knowledge about the disease and virus characteristics, such as its lethality and high transmissibility": I do not agree with the "lack of knowledge", it may be something else.

- p.2) It is important to distinguish two types of mobility: the flow of people between different areas (such as neighborhoods) and inside each of these areas. At the beginning of the epidemics, interventions of mobility between countries, or even municipalities are important to mitigate the spread from one place to the other by avoiding the mixing of people. Now, however, with cases confirmed in all municipalities and a high number of new cases and deaths every day, it is more important to use a social distancing approach and others NPIs, such as masks to reduce the level of contagion. The mobility can drive the spatio-temporal pattern, but other factors are more important to the local spread. In the way the mobility data was introduced in the model, it seems to be only related to the local spread.

- p.3) Lines 86-88: "shows" -> "show", "we concludes" -> "we conclude", "and discusses" -> "and discuss".

- p.3-4, and in the rest of the paper): Please choose a date format and keep that throughout the text. On page 3, the MM/DD/YYYY is used, then in line 135 the format MM/DD/YYYY is used together with DD/MM/YYYY (probably a typo). Later, in Fig 1c, MM/DD/YYYY is used, but in Figs 5(a-e), 6(a-f) and 10 the YYYY-MM-DD format is used.

- p.4) Is the quotation in lines 118-120 really necessary for the paper?

- p.5) The "Coronavirus Panel data" contains the number of cases and deaths by confirmation or report date, not the notification date as in "Opendata SUS". The first is affected by delays related to inserting the record in the system, the exam collection date, the exam result date, and finally the reporting from the municipality to the state's health department. So, the methodology used by Abbott et al is not enough, in this case, to "rewind" the data by using only the delays from symptoms onset to the notification. It is clear to me when I see Fig. 1(c) and 1(d): the peak around September 2020 appears as a sudden drop in Fig. 1(d). The peak is related to delays in confirming the cases, not in notifying the cases in the system (when the patient seeks medical attention), as the authors correctly stated in lines 178-179. However, I am not convinced that an adequate methodology was used in this case. Please see other methodologies to correct reporting delays, such as

- "A modelling approach for correcting reporting delays in disease surveillance data." Statistics in Medicine 38.22 (2019): 4363-4377.

- Fig 1: there are no labels in Figs. 1(a,b). What do the y-axis and x-axis mean? In Fig. 1(b), what does the dashed line mean? Why (a) and (b) are in different plots, if (a) is the empirical distribution of the delays and (b) the estimated one? Should not the bars be the empirical distribution and the curve the estimated one? Please clarify.

- p.5, 6, and so on) Another problem with the notation appears in the manuscript: $I_{N_i}$ and $I_{U_i}$ are used in p. 5, but in Eqs. (1-5) they appear as $I_{C_i}$ and $I_{S_i}$, and $I_{C_i}$ in line 266 (p. 8). Please clarify.

- p.7) $\\mu_i$ and $\\mu$ are not defined.

- p.7) "Alternatively, if this information is associated as a function of smaller regions, we can make inferences with finer granularity." What does that mean?

- p.7) Notation problem: here "$R_t$" is used, but in page 13 it appears as $R(t)$ (and caption of Fig. 7)

- p.7) The value of $q_t$ is calibrated in $W_{ij}(t) = q_t C_{ij}$ to find the values of $q_t$ that better represent the situation in a given time window, correct? The problem is that as $\\mu_i$ is a free parameter (and not defined), anything can "fit" the real data. Also, it is not clear if $C_{ij}$ is constant in time.

- p.7) The force of infection is $\\sum_j \\left [ W_{ij}(t) \\left( \\frac{I_{N_i}(t) + I_{U_i}(t)}{N_j} \\right) \\right]$, meaning that infected individuals from $i$ interact with $N_j$ individuals in $j$. What is the explanation for that choice? See Sec. 7.2 of

- Keeling, Matt J., and Pejman Rohani. Modeling infectious diseases in humans and animals. Princeton university press, 2011.

- p.7) In Eqs. (1-2) what is the meaning of the factor ${I_{C_i}(t)}/{N_i}$? Has this not already been counted in the infection force? Also, in Eqs. (1-2) it shows $\\lambda_{i,t}$, and later as $\\lambda_i(t)$.

- p.8) There is no information on the number of cases for each region, correct? I mean, the number of cases is only available at the city level, not by the regions.

- p. 9, Table 1) What are the values of $q_t$ for each case in Table 1? Maybe the median or mean value can be put in the table. The caption text does not describe anything and should be changed. Also, from where were these dates extracted for each city?

- p.9, line 298) There is a typo: "$q_1$ e $q_2$" -> "$q_1$ and $q_2$".

- p.10, Fig. 4) The caption says that $q_1$ and $q_2$ are being shown, but in Fig. 4(b) we have $q_3$. Please clarify.

- p.10) Were the results compared to a null model discarding the mobility data? I think it is important to state that the mobility data is necessary for the calibration. A calibration or sensitivity analysis is necessary to draw any statistical conclusion. My feeling is that the mobility is not playing a role here: a SEIR model with a time-dependent rate (related to the $q_t$ parameter in this case, and the dates in Table 1), without mobility data whatsoever, can be enough. See

- "A SEIR-like model with a time-dependent contagion factor describes the dynamics of the Covid-19 pandemic." medRxiv 2020.08.06.20169557 (2020)

- p.10, Fig. 5 and simulation results) The confidence interval is very weird. How do the authors explain these piece-wise confidence intervals? Were the data calibrated for each time window with a constant $q_t$, but a different set of other parameters? If so, that does not seem to be correct, since these time windows are not independent. See Fig. 5(c), for example. Why not plotting also the median or average value of the simulations, instead of only the weird confidence intervals?

- p.11, Fig. 6) The y-axis label seems to be wrong. For $q_0$, the value is close to 1. Is it the ratio between the number of infected cases reported using this strategy, compared to the calibrated one, or the %? In the caption, it says that it is the ratio, but uses "%" in the label. Please clarify.

- p.11 Fig. 6b) If it is the ratio, why are the values different of 1 when using $q_0$? What set of parameters was used in this case? Is it a fixed one, since no confidence interval is present?

- p.11 Fig. 6) Why some figures have $q_0$ and $q_2$, and others only $q_0$ in the legends? Please clarify.

- p.11) What the authors mean by "although it is likely to happen" in line 352? What about the new waves of infection with the new variants and mutations?

- p.11) "HDI" is not defined here, only on the next page.

- p.12 and Fig. 7) It was not clear to me if the number of infected people in each region of Fortaleza was obtained from the simulations or if they are real data. If so, what is the source of this data? If it is official reporting data, it does not seems to be related to any mobility data mentioned in the main text.

- p.13) The analysis of the correlation between DETRAN-CE and Google Mobility data is very interesting, especially the lag between $R(t)$ and the time series.

- p.14) I understand that "using a regional approach, we can stratify the information for each region individually". In the caption of Fig. 11, the authors state the number of infected cases was estimated by their model. Without comparison to official reporting data aggregated by region rather than the city as a whole, we cannot conclude anything. If the mobility data is used in a model, it must be compared to official data, or with a null model to show that this was really necessary. Also, again I stress the need for sensitivity analysis of the calibrated parameters.

- p. 14, Fig. 11) Why is Rt so large close to 04/06? No discussion was made about this fact.

- p. 15) My feeling is that similar conclusions for all municipalities investigated could be obtained without using the mobility data. Instead of using $W_{ij}$, the authors could use, for example, the demographic density to estimate the number of contacts of each region, multiplied by $q_t$ to infer the social distancing and others NPIs.

- p. 15, Fig. 12) Again, the caption says that it is the ratio, but the plot shows "# (Number) of". By the way, I suggest replacing all the "# of" with "Number of".

6. PLOS authors have the option to publish the peer review history of their article (what does this mean?). If published, this will include your full peer review and any attached files.

Reviewer #1: No

Reviewer #2: No

---

## [Author Response · Author response to Decision Letter 0]

16 Aug 2021

Thanks for handling this manuscript.

We have prepared a revised version taking into account all the comments and suggestions made by the reviewers.

We found all the reviews constructive, and we would like to thank the reviewers for helping us make a better contribution.

This response letter addresses all the comments in red, followed by

our responses, and, whenever necessary, the changes made (in black).

We also include the diff article between the prior and current versions, where deletions are in red, and additions are in blue.

---

## [Decision Letter · Decision Letter 1]

15 Sep 2021

PONE-D-21-13982R1Effects of  population mobility on the COVID-19 spread in BrazilPLOS ONE

Dear Dr. Ramos,

Thank you for submitting your manuscript to PLOS ONE. After careful consideration, we feel that it has merit but does not fully meet PLOS ONE’s publication criteria as it currently stands. Therefore, we invite you to submit a revised version of the manuscript that addresses the pertinent points raised by the reviewers.

We look forward to receiving your revised manuscript.

Kind regards,

Sergio Gómez

Academic Editor

PLOS ONE

Journal Requirements:

Reviewers' comments:

Reviewer's Responses to Questions

**Comments to the Author**

1. If the authors have adequately addressed your comments raised in a previous round of review and you feel that this manuscript is now acceptable for publication, you may indicate that here to bypass the “Comments to the Author” section, enter your conflict of interest statement in the “Confidential to Editor” section, and submit your "Accept" recommendation.

Reviewer #1: All comments have been addressed

Reviewer #2: (No Response)

2. Is the manuscript technically sound, and do the data support the conclusions?

Reviewer #1: Yes

Reviewer #2: Yes

3. Has the statistical analysis been performed appropriately and rigorously? 

Reviewer #1: Yes

Reviewer #2: Yes

4. Have the authors made all data underlying the findings in their manuscript fully available?

Reviewer #1: Yes

Reviewer #2: No

5. Is the manuscript presented in an intelligible fashion and written in standard English?

Reviewer #1: Yes

Reviewer #2: (No Response)

6. Review Comments to the Author

Reviewer #1: I am very satisfied with the new version of the manuscript, I was genuinely impressed by the effort the authors put in improving the overall quality of the work. I believe that the authors have addressed properly all the problems pointed out by the referees, therefore I am now happy to recommended it for publication in Plos One.

There are just a few more minor issues that I would invite the authors to consider before submitting the final version:

1) If I have understood correctly, the model is fitted to the cumulative number of cases, however it is a more common practice to fit it to the daily incidences, which is also a much more insightful variable. With that I am not saying that the authors should do the fit again, but at least adding a couple of panels in order to see if the fitted model captures daily incidences as well would be definitely a nice add.

2) Lines 408/409: "In particular, in Scenario I, we observed a delay in the peak occurrence, i.e., there is a prolongation of the effects caused by the pandemic." This sentence is confusing and risky, because the reader would expect to see an estimation of the difference in the *incidence* peak, not the peak of the ratio predicted_cases/observed_cases, which might not coincide with peak incidence. Also, the peak for this of metric appears only in scenario 1, while the trend in scenario 2 is decreasing overall, therefore is not very straightforward to see where the peak is. I suggest the authors to either drop or clarify this message of peak delay.

3) Fig. 1: I am happy the authors now specify that the vertical lines represent the mean values, but it would be nice to read the value of both as well, since the xscales are different, therefore is not easy to evaluate them. I suggest the authors to provide the values of the means (possibly standard deviation as well) in the caption.

4) Fig. 4: Are those the posteriors for q1 and q2 in Fortaleza? Or is it a different city? Please specify.

5) I think the caption of Fig.8 refers to Figure 9 and viceversa, please check.

6) Figure 9b: the line of DETRAN daily trajectories appears quite noisy, maybe the authors could to consider to add a moving averaged version for the sake of readability.

Reviewer #2: The paper has improved after revision and the authors addressed all my comments, but I am not entirely convinced that the mobility matrix itself is so important here. I do understand that the idea is to put it so that it is possible to implement restriction scenarios, but this can be done with a time-dependent factor, such as $q_t$ but not bounded by [0,1], as shown in the response letter. So, I wonder what happens to the results if, for example, the matrix elements are now random, keeping the total number of edges, weights, and nodes of the network; please see some network randomization procedures. Also, I still find necessary some revisions on the presentation of the figures so that they have the same date format (some are YYYY-MM-DD and others MM/DD, some tick labels are rotated, others not). By the way, there is a Ref. that can be interesting since the paper deals with data from Ceará: "Superspreading k-cores at the center of COVID-19 pandemic persistence", arXiv:2103.08685 (2021).

7. PLOS authors have the option to publish the peer review history of their article (what does this mean?). If published, this will include your full peer review and any attached files.

Reviewer #1: No

Reviewer #2: No

---

## [Author Response · Author response to Decision Letter 1]

29 Oct 2021

We attached a document which details all responses to reviewers' comments.

---

## [Decision Letter · Decision Letter 2]

15 Nov 2021

Effects of  population mobility on the COVID-19 spread in Brazil

PONE-D-21-13982R2

Dear Dr. Ramos,

We’re pleased to inform you that your manuscript has been judged scientifically suitable for publication and will be formally accepted for publication once it meets all outstanding technical requirements.

Kind regards,

Sergio Gómez

Academic Editor

PLOS ONE

Additional Editor Comments (optional):

Reviewers' comments:

Reviewer's Responses to Questions

**Comments to the Author**

1. If the authors have adequately addressed your comments raised in a previous round of review and you feel that this manuscript is now acceptable for publication, you may indicate that here to bypass the “Comments to the Author” section, enter your conflict of interest statement in the “Confidential to Editor” section, and submit your "Accept" recommendation.

Reviewer #2: All comments have been addressed

2. Is the manuscript technically sound, and do the data support the conclusions?

Reviewer #2: (No Response)

3. Has the statistical analysis been performed appropriately and rigorously? 

Reviewer #2: (No Response)

4. Have the authors made all data underlying the findings in their manuscript fully available?

Reviewer #2: (No Response)

5. Is the manuscript presented in an intelligible fashion and written in standard English?

Reviewer #2: (No Response)

6. Review Comments to the Author

Reviewer #2: Please check the label of Fig. 6 with "values to be predicted" instead of "predicted values". No further comments.

7. PLOS authors have the option to publish the peer review history of their article (what does this mean?). If published, this will include your full peer review and any attached files.

Reviewer #2: No

---

## [Editor Report · Acceptance letter]

17 Nov 2021

PONE-D-21-13982R2 

Effects of Population Mobility on the COVID-19 Spread in Brazil 

Dear Dr. Ramos:

I'm pleased to inform you that your manuscript has been deemed suitable for publication in PLOS ONE. Congratulations! Your manuscript is now with our production department. 

Kind regards, 

on behalf of

Dr. Sergio Gómez 

Academic Editor

PLOS ONE